# Prompt Reinjection: Alleviating Prompt Forgetting in Multimodal Diffusion Transformers

**Yuxuan Yao** [* 1 2] **Yuxuan Chen** [* 1] **Hui Li** [1] **Kaihui Cheng** [1] **Qipeng Guo** [3] **Yuwei Sun** [4] **Zilong Dong** [5]
**Jingdong Wang** [6] **Siyu Zhu** [1 2]

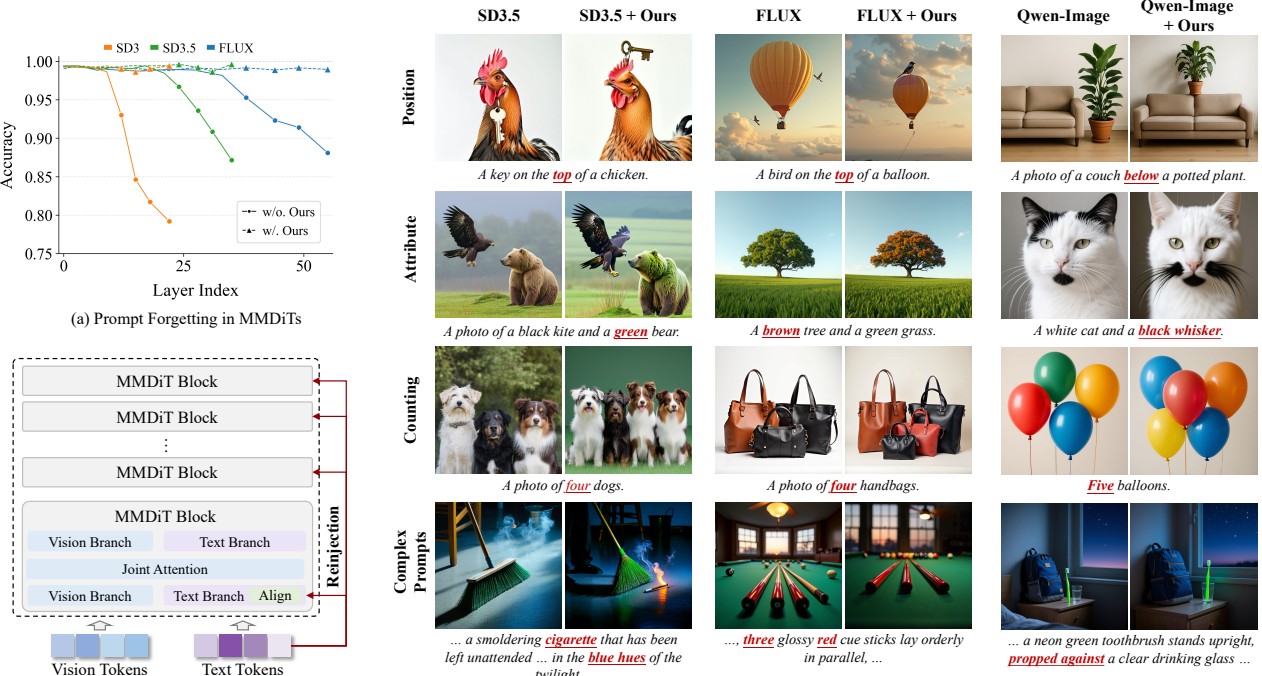

Figure 1. **Prompt forgetting in MMDiTs and Prompt Reinjection.** (a) We quantify *prompt forgetting* by probing token-level attribute recoverability. Accuracy drops monotonically with depth in SD3, SD3.5, and FLUX, indicating progressive loss of fine-grained prompt information in deeper text features. (b) We propose *Prompt Reinjection*: reinjecting aligned shallow-layer text features into later blocks during inference. (a) With Prompt Reinjection enabled, probing accuracy remains stable across depth, showing effective mitigation of forgetting. (c) Prompt Reinjection improves instruction following across multiple MMDiT variants, more consistently satisfying prompt constraints under diverse prompt styles.

## Abstract

Multimodal Diffusion Transformers (MMDiTs) for text-to-image generation maintain separate text and image branches, with bidirectional information flow between text tokens and visual latents throughout denoising. In this setting, we observe a *prompt forgetting* phenomenon: the semantics of the prompt representation in the text branch is progressively forgotten as depth increases. We further verify this effect on three representative MMDiTs—SD3, SD3.5, and FLUX.1 by probing linguistic attributes of the representations over the layers in the text branch. Motivated by these findings, we introduce a training-free approach, *prompt reinjection*, which reinjects prompt representations from early layers into later layers to alleviate this forgetting. Experiments on GenEval, DPG, and T2I-CompBench++ show consistent gains in instruction-following capability, along with improvements on metrics capturing preference, aesthetics, and overall text–image generation quality. Our code is available at *https://github.com/fudan-generative-vision/PromptReinjectio*n.

[1]Fudan University, China [2]Shanghai Innovative Institute, China [3]Shanghai AI Laboratory, China [4]Shanghai Academy of AI for Science, China [5]Alibaba Group, China [6]Baidu, China. Correspondence to: Siyu Zhu <siyuzhu@fudan.edu.cn>.

*Proceedings of the 43rd International Conference on Machine Learning*, Seoul, South Korea. PMLR 306, 2026. Copyright 2026 by the author(s).

# 1. Introduction

Text-to-image generation has witnessed remarkable advancements, with diffusion models emerging as the dominant framework for high-fidelity and controllable generation (Ho et al., 2020; Nichol and Dhariwal, 2021; Dhariwal and Nichol, 2021; Ho and Salimans, 2022). Previous approaches typically condition a U-Net denoiser via cross-attention within a learned latent space, an architecture that facilitates both computational efficiency and scalable prompt-driven generation (Rombach et al., 2022b; Nichol et al., 2021; Saharia et al., 2022; Podell et al., 2023; Betker et al., 2023). More recently, Diffusion Transformers (DiTs) have demonstrated superior scalability and representational capacity for complex compositions while continuing to utilize cross-attention for text injection. This shift highlights the increasing importance of in-context conditioning for adhering to complicated instructions and multi-modal conditioning (Peebles and Xie, 2023; Chen et al., 2023; 2024a; Xie et al., 2024; Zheng et al., 2023).

Instead of treating text as a fixed (as network depth increases) external condition injected into an image denoiser, *multimodal diffusion transformers (MMDiTs)* jointly process textual and visual latent tokens within a unified transformer stack, facilitating bidirectional interaction throughout the denoising process. Prominent architectures, such as Stable Diffusion 3 (SD3) (Esser et al., 2024) and its successors, leverage this joint processing to improve the handling of complex prompts (Esser et al., 2024; Labs, 2024; Wu et al., 2025; Cai et al., 2025). The defining characteristic of these models is the iterative transformation of both modalities: text representations evolve alongside visual latents layer-by-layer.

While this unified evolution potentially strengthens cross-modal coupling, it also implies that text features are iteratively transformed at every layer rather than serving as a fixed conditioning anchor. In current MMDiT architectures, text tokens evolve alongside visual latents; however, the diffusion objective remains localized to the visual latent space (e.g., $\epsilon$-, $x_0$-, or $v$-prediction) (Ho et al., 2020; Rombach et al., 2022b). Consequently, visual tokens receive direct supervision, whereas textual representations are updated only indirectly via their influence on visual reconstruction through joint attention. This supervisory asymmetry imposes minimal constraints on the semantic preservation of text features; under successive transformer blocks, the model may minimize denoising error without preserving fine-grained prompt semantics. As a result, intermediate text representations undergo significant drift in deeper layers, leading to what we term *Prompt Forgetting*–a phenomenon where token-level textual information becomes progressively unrecoverable.

To empirically characterize this phenomenon, we perform a systematic layer-wise analysis of intermediate text features across several representative MMDiT architectures, including SD3 (Esser et al., 2024), SD3.5 (Esser et al., 2024), FLUX (Labs, 2024), and Qwen-Image (Wu et al., 2025). Our investigation proceeds in two stages: (i) an observational analysis where we measure the preservation of *local semantic structure* via Conditional K-Nearest Neighbor Alignment (CKNNA) (Huh et al., 2024) and visualize global distributional drift within a shared PCA space (Sec. 4.1); and (ii) a functional quantification via layer-wise recoverability probes. In the latter, we train lightweight classifiers to decode token-level linguistic attributes from intermediate representations (Sec. 4.2). Our results across all models reveal a consistent depth-wise degradation in semantic preservation, pronounced distributional shifts, and a monotonic decline in probing accuracy. These findings provide rigorous evidence that token-level prompt information becomes progressively unrecoverable in deeper layers, confirming the emergence of prompt forgetting.

Building on these insights, we propose *Prompt Reinjection*, a training-free, inference-time intervention (Sec. 5) that mitigates semantic loss by reintroducing aligned shallow-layer text signals into deeper transformer blocks. Systematic evaluations across GenEval (Ghosh et al., 2023), DPG-Bench (Hu et al., 2024), and T2I-CompBench++ (Huang et al., 2025) demonstrate robust improvements in instruction following across a range of generation tasks, including attribute binding (e.g., color/shape/texture), numeracy, multi-object composition, and spatial relations. On GenEval, Prompt Reinjection improves the overall scores of SD3.5 and FLUX by 6.48% and 5.64%, respectively. We also report consistent gains across broader quality dimensions, spanning human preference (via HPSv2 (Wu et al., 2023), ImageReward-v1 (Xu et al., 2023) and PickScore (Kirstain et al., 2023)) , and global semantic alignment (via CLIP score (Radford et al., 2021)). Specifically, HPSv2 is evaluated on GenEval samples, while the remaining metrics are reported on the COCO-5K dataset (Lin et al., 2014).

# 2. Related Work

**Diffusion Transformers and Multimodal Architectures.**

Diffusion models (Ho et al., 2020; Song et al., 2020) have become the dominant approach for natural language-driven high-resolution image generation (Wei et al., 2023; Chen et al., 2023; Rombach et al., 2022a; Ma et al., 2024; Wan et al., 2025). Most advances rely on effective ways to inject text conditions, improving cross-modal alignment and generation quality. Early models such as SD1.5 (Rombach et al., 2022a), SDXL (Podell et al., 2023), and Imagen (Saharia et al., 2022) follow a decoupled "U-Net + text cross-attention" design, where text is injected into the denoiser as an external condition, but this separation limits deeper modality coupling. DiT (Peebles and Xie, 2023) replaces the U-Net denoiser with a Transformer backbone and motivates

DiT-style text-to-image models that incorporate prompts via standard conditioning modules (e.g., cross-attention or modulation) (Chen et al., 2023; 2024a; Zhou et al., 2025; Xie et al., 2024). Building on this trend, Multimodal Diffusion Transformers (MMDiTs) further process text tokens and visual latent tokens *together* inside the denoising stack, enabling bidirectional interaction through joint attention. Representative models include SD3 (Esser et al., 2024), FLUX (Labs, 2024), and Qwen-Image (Wu et al., 2025), which share this core MMDiT design while differing in architectural choices (e.g., token mixing strategies or prompt encoders).

**Text-to-Image Alignment and Instruction Following.** Text–image alignment and instruction following are central goals of text-driven visual generation. Prior work improves these capabilities through training-free interventions, explicit layout control, feedback-driven optimization, or attention modulation (Chen et al., 2024b; Black et al., 2023; Dahary et al., 2024; Fan et al., 2023; Chefer et al., 2023; Rassin et al., 2023). However, most of these are designed around U-Net denoisers and do not transfer directly to the DiT-style diffusion models (Peebles and Xie, 2023). Recent work has started to analyze MMDiTs at the component level (Avrahami et al., 2025; Wei et al., 2025; Shin et al., 2025), but it has not directly characterized the layerwise evolution of intermediate text features during denoising. Several concurrent studies connect text-feature flow to instruction following capability: TACA (Lv et al., 2025) analyzes how the imbalance between text and image tokens can suppress cross-modal attention, and introduces a timestep-aware reweighting mechanism to better preserve text conditioning; (Li et al., 2026) studies how text features from different layers affect different facets of generation, and amplifies selected-layer text features to strengthen their conditioning effect.

# 3. Preliminaries

## 3.1. Multimodal Diffusion Transformers (MMDiTs)

The MMDiT architecture unifies the processing of textual and visual modalities within a shared transformer backbone. Formally, let a tokenized prompt be encoded into a sequence of text embeddings $T^{(0)} \in \mathbb{R}^{n \times d}$, and an image be represented in the latent space as a sequence of tokens $I^{(0)} \in \mathbb{R}^{m \times d}$, where $n$ and $m$ denote the respective sequence lengths and $d$ represents the hidden dimension.

At each layer $l$, the model maintains modality-specific features $T^{(l)}$ and $I^{(l)}$. These are concatenated along the sequence dimension to form a joint multimodal hidden state:

$$Z^{(l)} = [T^{(l)}; I^{(l)}] \in \mathbb{R}^{(n+m) \times d} \quad (1)$$

Unlike traditional cross-attention architectures that condition visual features on a static text representation, MMDiT performs a unified self-attention operation over the entire sequence $Z^{(l)}$. To account for domain discrepancies between text and vision, the architecture typically employs modality-specific linear projections for queries, keys, and values. The update rule for a single layer is defined as:

$$Z^{(l+1)} = \text{TransBlock}(Z^{(l)}; \Theta^{(l)}) \quad (2)$$

where $\Theta^{(l)}$ denotes the layer-specific parameters. This design facilitates bidirectional cross-modal interaction: visual latents are conditioned on textual context, while text representations are iteratively refined based on the evolving visual features.

## 3.2. Training of MMDiTs and Supervision Imbalance

The denoiser $f_\theta$ is optimized within a latent diffusion framework to minimize the discrepancy between added and predicted noise. Given a clean latent $x_0$, a noise vector $\epsilon \sim \mathcal{N}(0, \mathbf{I})$, and a prompt condition $c$, the standard $\epsilon$-prediction objective is:

$$\mathcal{L}_\epsilon = \mathbb{E}_{t, x_0, \epsilon} \left[ \| f_\theta(z_t, t, c) - \epsilon \|_2^2 \right] \quad (3)$$

where $z_t$ is the noisy latent at timestep $t$. While alternative parameterizations such as $x_0$ or $v$-prediction are common, they maintain a fundamental modality supervision imbalance.

Because the loss function is defined exclusively within the image latent space, visual tokens receive direct supervision. In contrast, gradients for textual tokens $T^{(l)}$ are only propagated indirectly via the unified attention mechanism:

$$\nabla_{T^{(l)}} \mathcal{L}_\epsilon = \frac{\partial \mathcal{L}_\epsilon}{\partial I^{(L)}} \cdot \frac{\partial I^{(L)}}{\partial T^{(l)}} \quad (4)$$

This supervisory asymmetry implies that the optimization process imposes minimal constraints on the semantic preservation of text features, provided the representations remain sufficiently informative for the immediate denoising task.

Consequently, intermediate text features $T^{(l)}$ can change substantially across layers—such as weakening the preservation of *local semantic structure* (relative to the text-encoder space) and inducing large *distributional shifts* in the token-feature space—which may precipitate the loss of finegrained linguistic attributes as depth increases, a phenomenon we formalize as *Prompt Forgetting*.

# 4. Prompt Forgetting in MMDiTs

MMDiT-based text-to-image models jointly process textual and visual latents within a unified stack, yet the denoising objective exclusively supervises visual predictions. As detailed in Sec. 3.2, this supervisory asymmetry raises a fundamental question: *absent explicit semantic constraints, do intermediate text features progressively discard prompt-related information as depth increases?*

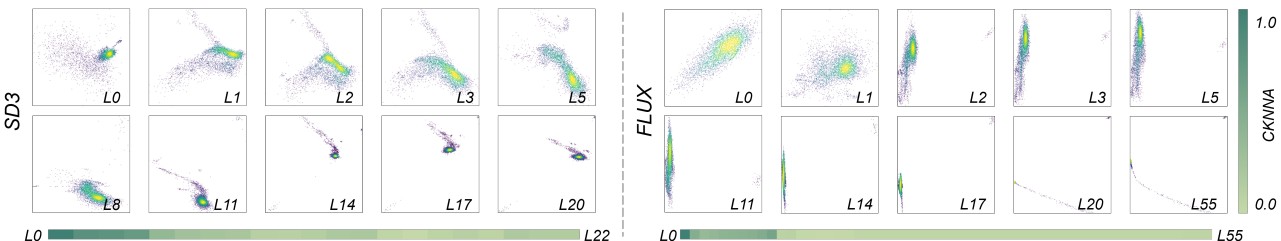

*Figure 2.* Overall observation of per-layer text-token representations in SD3-medium and FLUX.1-Dev.

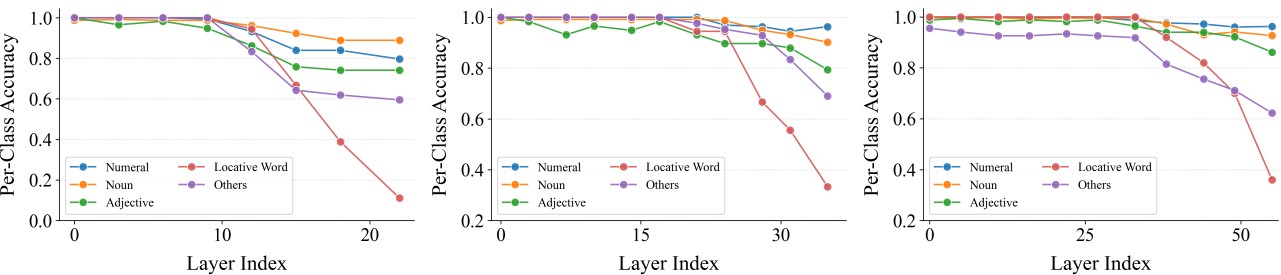

*(a)* Per-category accuracy for SD3-medium.    *(b)* Per-category accuracy for SD3.5-large.    *(c)* Per-category accuracy for FLUX.1-Dev.

*Figure 3.* Probe accuracy reveals *prompt forgetting* in MMDiT text features. Each subplot reports per-category test accuracy when decoding token-level attributes from intermediate text representations at each layer for SD3-medium (left), SD3.5-large (middle), and FLUX.1-Dev (right).

We investigate this phenomenon via a two-stage analytical framework. First, we perform an observational analysis (Sec. 4.1) to characterize the evolution of text representations through the lenses of local semantic structure and global distributional drift. Second, we provide a functional quantification of information loss (Sec. 4.2) using layer-wise recoverability probes. Here, we operationalize Prompt Information through the lens of recoverability, defining a token-level attribute as recoverable if it can be reliably decoded from its intermediate representation.

Our study encompasses several prominent MMDiT architectures, including SD3-medium (Esser et al., 2024), SD3.5-large (Esser et al., 2024), and FLUX.1-dev (Labs, 2024).

### 4.1. Layer-wise Text Representation Drift

We first characterize how text-token representations transform across depth in MMDiTs. Specifically, we adopt two complementary analytical perspectives: Conditional K-Nearest Neighbor Alignment (CKNNA) (Huh et al., 2024) to quantify the preservation of local semantic structures, and shared PCA projections to visualize global distributional shifts of text tokens across the transformer stack.

**Semantic Observation:** To test whether input-level semantic neighborhoods are preserved across depth, we compute CKNNA between $T^{(l)}$ and $T^{(0)}$ (Appendix B.1). Concretely, for each text token, we retrieve its $k$ nearest neighbors in the reference space $T^{(0)}$ and in the layer-$l$ space $T^{(l)}$, and measure how often the neighbor identities are preserved. CKNNA can be written as the average overlap of $k$-NN sets:

$$\text{CKNNA}(l) \;=\; \frac{1}{N} \sum_{i=1}^{N} \frac{\left| \mathcal{N}_k^{\text{in}}(i) \cap \mathcal{N}_k^{(l)}(i) \right|}{k}, \quad (5)$$

where $\mathcal{N}_k^{\text{in}}(i)$ and $\mathcal{N}_k^{(l)}(i)$ denote the $k$-NN index sets of the $i$-th token in $T^{(0)}$ and $T^{(l)}$, respectively. A diminishing CKNNA score indicates that tokens sharing local semantic similarity at the input stage progressively diverge in deeper layers. Empirically, we observe a monotonic decline in CKNNA across all models (Fig. 2), suggesting progressively weaker preservation of local semantic structure.

**Distribution Observation:** Furthermore, we visualize the global trajectory of text features by projecting representations from all layers into a shared PCA space (Fig. 2). In both models, the majority of tokens progressively collapse into a highly concentrated region of the latent space, while only a sparse subset of outliers maintains distinct separation. This concentration indicates that token features become less spread out and potentially less separable, and may undermine the recoverability of fine-grained prompt information in deeper text features.

Together, these observations provide a coherent picture of layer-wise text-representation changes in MMDiT, which motivates our next step: directly testing whether token-level prompt information becomes less recoverable at deeper layers (Sec. 4.2).

### 4.2. Layer-wise Text Information Probing

We quantify the *recoverability* of token-level attributes via a supervised token-category probing task. Let $y_i$ denote

the semantic category of the $i$-th token, and $t_i^{(l)} \in \mathbb{R}^d$ its representation at layer $l$. We train layer-specific lightweight MLP classifiers $g_l : \mathbb{R}^d \to \{1, \dots, C\}$ to predict $y_i$ from $t_i^{(l)}$, defining recoverability as the test set accuracy:

$$\text{Rec}(l) \;=\; \mathbb{E}\big[\mathbb{I}\big(g_l(t_i^{(l)}) = y_i\big)\big]. \quad (6)$$

Crucially, as all probes utilize identical architectures and training protocols, variations in $\text{Rec}(l)$ directly reflect the degradation of accessible token-level information within the representations at each depth.

We curate a labeled dataset from GenEval (Ghosh et al., 2023) prompts, annotating tokens into five linguistic categories (*noun*, *adjective*, *spatial-relation*, *numeral*, *others*) and propagating labels to sub-tokens. For each architecture, we extract features from a single denoising step to train layer-wise probes, using the text encoder output (Layer 0) as the baseline (details in Appendix B.2). As illustrated in Fig. 1 (a), overall probing accuracy exhibits a monotonic decline across layers for both SD3, SD3.5 and FLUX Given the controlled probe capacity, this trend provides rigorous quantitative evidence that fine-grained textual information becomes progressively unrecoverable in deeper layers.

A fine-grained analysis reveals that this information loss is non-uniform across linguistic categories (Fig. 3). Notably, spatial-relation tokens suffer the most precipitous accuracy drop, indicating that positional semantics are discarded more aggressively than object or attribute information. This pattern aligns with our instruction-following evaluation (Table 1), where these models perform worst on spatial reasoning tasks.

# 5. Alleviating Prompt Forgetting

To alleviate this forgetting phenomenon identified in Sec. 4, we propose **Prompt Reinjection**, a training-free, inference-time intervention designed to mitigate this semantic loss. By reintroducing high-fidelity prompt signals from shallow layers into deeper transformer blocks via residual connections, *Prompt Reinjection* enhances the model's instruction-following capabilities without requiring parameter updates. The proposed framework relies on two distinct phases. First, in Sec. 5.1, we validate the semantic fidelity of shallow features and the feasibility of residual injection through a pilot study. Second, in Sec. 5.2, we formalize the Prompt Reinjection mechanism, which addresses the distributional and geometric mismatches across layers via statistical anchoring and orthogonal Procrustes alignment.

## 5.1. Pilot Study: Semantic Fidelity of Shallow Reinjection

Before formalizing *Prompt Reinjection*, we conduct a pilot study to verify two prerequisites: (i) whether shallow text features $T^{(0)}$ retain accessible prompt semantics, and (ii)

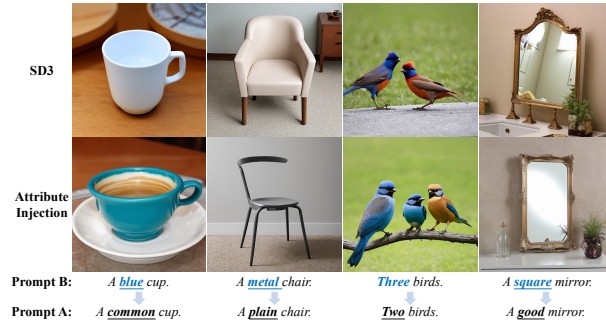

| | | | | |
|---|---|---|---|---|
| **Prompt B:** | *A **blue** cup.* | *A **metal** chair.* | ***Three** birds.* | *A **square** mirror.* |
| **Prompt A:** | *A **common** cup.* | *A **plain** chair.* | ***Two** birds.* | *A **good** mirror.* |

*Figure 4.* Residual attribute injection results. During generation with prompt $A$, injecting shallow text features from prompt $B$ steers outputs toward the injected attribute, indicating that shallow residuals carry transferable semantics.

whether residual addition serves as a viable mechanism for semantic transfer. We construct *minimal pair* prompts $(P_A, P_B)$ of identical length, where $P_B$ modifies a single attribute of $P_A$. During the denoising process of $P_A$, we inject a scaled residual of the shallow features from $P_B$ into the deeper blocks of the MMDiT:

$$T_A^{(l)} \leftarrow T_A^{(l)} + w \cdot T_B^{(0)}, \quad l \geq 2, \quad w \in [0.01, 0.1] \quad (7)$$

where $T_A^{(l)}$ denotes the text features of prompt $A$ at layer $l$. As illustrated in Fig. 4, the generated images shift consistently toward the attributes defined in $P_B$ (e.g., specific material or quantity changes).

## 5.2. Prompt Reinjection

Although residual injection facilitates semantic transfer, naive cross-layer addition is often hindered by significant distributional (scale and shift) and geometric (coordinate system) discrepancies between features at different depths. To ensure stable and effective fusion, our *Prompt Reinjection)* integrates two mechanisms: (1) **Distribution Anchoring**, which normalizes the fusion space and restores target statistics, and (2) **Geometry Alignment**, which maps origin features to the target manifold via an orthogonal Procrustes transform. Let $T_{\text{ori}}, T_{\text{tgt}} \in \mathbb{R}^{n \times d}$ denote the text-token features at the origin and target layers, respectively.

**Distribution Anchoring and Restoration.** To resolve discrepancies in feature magnitude and offset, we perform semantic fusion within a standardized latent space. For target features $T_{\text{tgt}}$, we compute the token-wise mean $\mu_{\text{tgt}}$ and standard deviation $\sigma_{\text{tgt}}$. Prior to injection, we apply Layer Normalization (LN) to homogenize both representations:

$$\hat{T}_{\text{ori}} = \text{LN}(T_{\text{ori}}), \quad \hat{T}_{\text{tgt}} = \text{LN}(T_{\text{tgt}}) \quad (8)$$

Following the fusion step (Eq. 11), we project the augmented features $T_{\text{added}}$ back to the original statistical distribution of the target layer:

$$T_{\text{final}} = T_{\text{added}} \odot \sigma_{\text{tgt}} + \mu_{\text{tgt}} \quad (9)$$

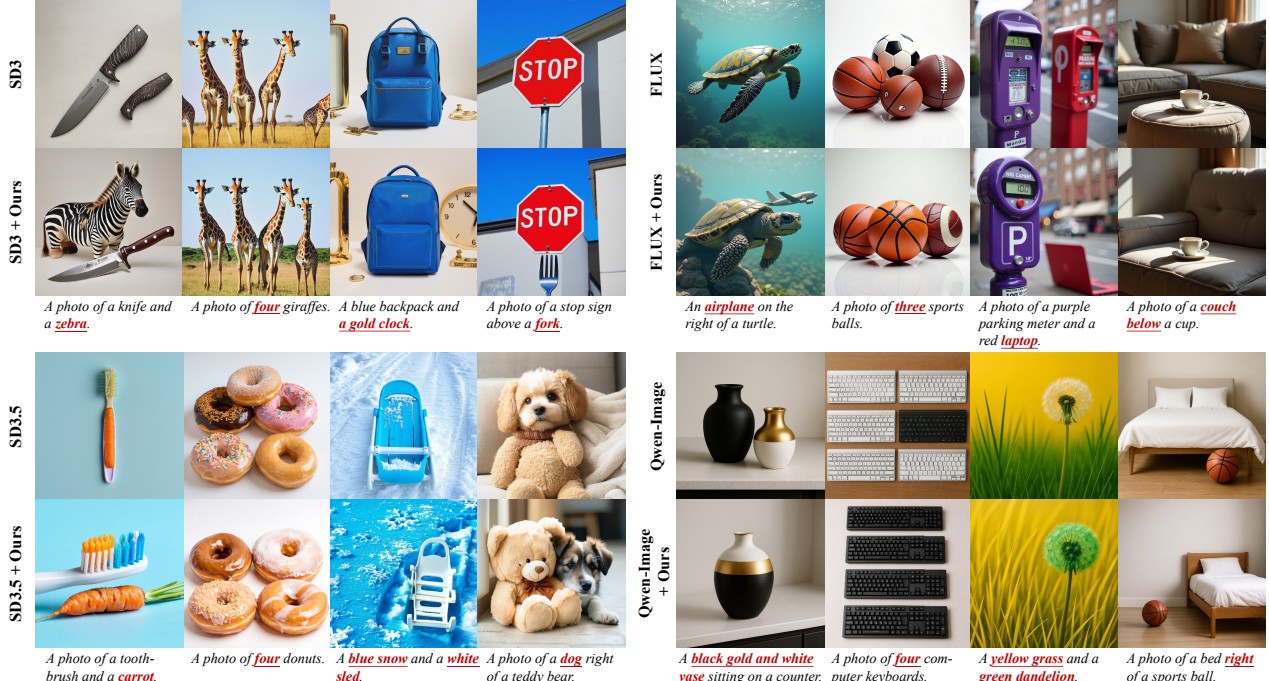

*Figure 5.* **Qualitative** comparison between each base model (SD3-medium, SD3.5-large, FLUX.1-Dev, and Qwen-Image) and its counterpart with Prompt Reinjection enabled. Bold text in the prompts highlights the constraints where our method improves text–image consistency over the base models.

This anchoring mechanism ensures the modified representations remain within the numerical range expected by subsequent transformer blocks, preserving generative stability.

**Geometry Alignment via Orthogonal Procrustes.** While normalization reconciles first- and second-order statistics, it does not correct for the rotation of latent coordinate systems across depths. We address this via an orthogonal Procrustes transformation. During a one-time calibration phase using COCO-5K, we extract normalized text features $\mathbf{X}, \mathbf{Y} \in \mathbb{R}^{N \times d}$ from the origin and target layers, respectively. We solve for the optimal orthogonal rotation $R$ that minimizes reconstruction error:

$$\min_R \|\mathbf{X}R - \mathbf{Y}\|_F^2 \quad \text{s.t. } R^\top R = I \tag{10}$$

Using Singular Value Decomposition (SVD), where $U\Sigma V^\top = \text{SVD}(\mathbf{X}^\top \mathbf{Y})$, the closed-form solution is given by $R = UV^\top$. At inference time, the origin features are aligned and injected via:

$$T_{\text{added}} = \hat{T}_{\text{tgt}} + w \cdot \hat{T}_{\text{ori}} R \tag{11}$$

where $w$ is a hyperparameter controlling injection strength.

**Selecting Origin and Target-layers.** The origin layer $l_{\text{ori}}$ is selected to balance semantic fidelity and cross-layer compatibility. While using the text-encoder output ($l = 0$) can already yield improvements, our PCA analysis (Fig. 2) shows that the first few MMDiT blocks undergo a sharp distributional transition when entering the denoiser. Choosing $l_{\text{ori}}$ as the shallowest layer after this transition reduces

the overall distribution and geometry gap between origin and target features, which makes subsequent injection more stable and typically more effective.

Since probing results in Fig. 1 shows an approximately monotonic depth-wise degradation of recoverability $\text{Rec}(l)$, we adopt a unified setting that injects into all deeper blocks after the origin, $L_{\text{tgt}} = \,l \mid l > l_{\text{ori}}\,$, which directly targets the depth range where $\text{Rec}(l)$ degrades. In Sec. 6.3, we further ablate layer choices and the injection weight $w$, and provide a more detailed analysis of how these design decisions affect performance.

## 6. Experiments

### 6.1. Experiment Settings

**Baselines and Benchmarks.** We evaluate our method on four representative MMDiT-based text-to-image models: SD3-medium (Esser et al., 2024), SD3.5-large (Esser et al., 2024), FLUX.1-dev (Labs, 2024), and Qwen-Image (Wu et al., 2025). All evaluations are conducted at a resolution of $1024 \times 1024$. For instruction following and text–image alignment, we report results on three widely used benchmarks: GenEval (Ghosh et al., 2023), DPG-bench (Hu et al., 2024), and T2I CompBench++ (Huang et al., 2025). As complementary quality signals, we report three widely used human-preference proxies—HPSv2 (Wu et al., 2023) on GenEval generations, and ImageReward-v1 (Xu et al., 2023) and PickScore (Kirstain et al., 2023) on COCO-5K (Lin et al., 2014) generations—together with a CLIP-

*Table 1.* **Quantitative** comparison on **GenEval** (Ghosh et al., 2023) comparison between each *base model* (SD3-medium, SD3.5-large, FLUX.1-Dev, Qwen-Image) and the same model with our method enabled. Cells highlighted in light red indicate the better score within each base/augmented model pair.

| Model | GenEval | | | | | | |
|---|---|---|---|---|---|---|---|
| | **Overall** | Single obj. | Two obj. | Counting | Colors | Color attr | Position |
| SD3 | 0.6793 | 1.0000 | 0.8460 | 0.6031 | 0.8617 | 0.5500 | 0.2150 |
| + Ours | 0.7059 | 1.0000 | 0.8864 | 0.6375 | 0.9043 | 0.5575 | 0.2500 |
| SD3.5 | 0.7179 | 0.9969 | 0.9167 | 0.7062 | 0.8351 | 0.5950 | 0.2575 |
| + Ours | 0.7644 | 1.0000 | 0.9520 | 0.7344 | 0.9149 | 0.6650 | 0.3200 |
| FLUX | 0.6613 | 0.9875 | 0.8308 | 0.7281 | 0.7686 | 0.4575 | 0.1950 |
| + Ours | 0.6986 | 0.9969 | 0.8485 | 0.7500 | 0.8165 | 0.5275 | 0.2525 |
| Qwen Image | 0.8756 | 0.9844 | 0.9444 | 0.9062 | 0.8963 | 0.7675 | 0.7550 |
| + Ours | 0.8933 | 0.9875 | 0.9596 | 0.9156 | 0.9096 | 0.7850 | 0.8025 |

*Table 2.* **Quantitative** comparison of **automatic metrics** for each *base model* with and without our method. HPSv2 is measured on GenEval generations, while ImageReward, PickScore, and CLIP are measured on COCO-5K generations.

| Model | HPSv2 | ImageReward | PickScore | CLIP |
|---|---|---|---|---|
| SD3 | 0.2935 | 0.9514 | 22.57 | 0.2633 |
| + Ours | 0.2941 | 1.0825 | 22.60 | 0.2676 |
| SD3.5 | 0.2951 | 1.0574 | 22.61 | 0.2677 |
| + Ours | 0.2995 | 1.1565 | 22.73 | 0.2716 |
| FLUX | 0.2970 | 1.0695 | 23.05 | 0.2590 |
| + Ours | 0.2995 | 1.0811 | 22.97 | 0.2611 |
| Qwen Image | 0.3014 | 1.2721 | 23.17 | 0.2715 |
| + Ours | 0.3056 | 1.3077 | 23.38 | 0.2704 |

*Table 3.* **Quantitative** results on **DPG** (Hu et al., 2024) and **T2I-CompBench++** (Huang et al., 2025), comparing each *base model* to the same model with our method enabled. Cells highlighted in light red indicate the better score within each model pair.

| Model | DPG | | | | | | T2I-CompBench++ | | | | | | |
|---|---|---|---|---|---|---|---|---|---|---|---|---|---|
| | **Overall** | Entity | Attribute | Other | Relation | Global | Amount | Color | Shape | Texture | 2D-Spatial | 3D-Spatial | Non-Spatial |
| SD3 | 85.43 | 90.71 | 88.24 | 85.60 | 93.35 | 86.32 | 0.6050 | 0.7929 | 0.5704 | 0.7092 | 0.2913 | 0.4061 | 0.3114 |
| + Ours | 87.12 | 92.28 | 89.44 | 87.20 | 94.39 | 86.34 | 0.6056 | 0.8416 | 0.5965 | 0.7496 | 0.3044 | 0.4147 | 0.3140 |
| SD3.5 | 83.64 | 89.79 | 87.93 | 82.00 | 92.80 | 82.98 | 0.6368 | 0.7809 | 0.6084 | 0.7296 | 0.2894 | 0.3858 | 0.3179 |
| + Ours | 86.99 | 91.90 | 90.09 | 85.60 | 94.31 | 83.89 | 0.6398 | 0.8320 | 0.6532 | 0.7789 | 0.3003 | 0.3967 | 0.3197 |
| FLUX | 83.97 | 90.29 | 87.07 | 82.40 | 93.01 | 83.97 | 0.6134 | 0.7546 | 0.5086 | 0.6307 | 0.2765 | 0.4036 | 0.3069 |
| + Ours | 84.58 | 90.40 | 87.89 | 82.80 | 93.34 | 84.55 | 0.6292 | 0.7866 | 0.5172 | 0.6493 | 0.2779 | 0.4095 | 0.3079 |
| Qwen Image | 88.90 | 93.72 | 91.01 | 87.60 | 94.99 | 84.38 | 0.7540 | 0.8419 | 0.5978 | 0.7487 | 0.4536 | 0.4575 | 0.3163 |
| + Ours | 89.15 | 93.87 | 91.01 | 87.68 | 95.29 | 85.87 | 0.7764 | 0.8508 | 0.6138 | 0.7584 | 0.4594 | 0.4587 | 0.3166 |

based score (Zhengwentai, 2023) that measures global text–image semantic alignment on COCO-5K. Regarding comparison methods, we include TACA (Lv et al., 2025) in Appendix F, which applies timestep-aware reweighting to textual conditions in attention for MMDiTs.

**Implementation Details.** For each model, we follow the official default inference configuration, using the recommended CFG scale and number of sampling steps. We use the same random seeds for the baseline and our method to ensure fair comparisons. All experiments are conducted on a single NVIDIA H200 GPU. For each model, all reported results use the same inference configuration and reinjection setup. Detailed settings are provided in Appendix C.

### 6.2. Main Results

**Instruction Following Improvements.** Tables 1 and 3 show that enabling our method consistently improves instruction following and text–image alignment across four mainstream MMDiT-based models (SD3, SD3.5, FLUX, and Qwen-Image) on three standard benchmarks: GenEval, DPG-bench, and T2I-CompBench++. Notably, the improvements are non-uniform across sub-tasks, including color/attribute binding, numeracy, multi-object composition, and relation understanding, are highly correlated with our probing findings. In particular, Position tasks in GenEval show the clearest and most consistent improvements across models (Table 1). This empirically validates our hypothesis: since spatial semantics undergo the most severe degradation (as shown in Fig. 3 ), explicitly reinjecting shallow features yields the most significant correction in spatial adherence.

Across models, the magnitude of improvement varies with the base capability. For the 20B-parameter Qwen-Image, while performance on simpler attributes (e.g., colors) approaches metric saturation, Prompt Reinjection still demonstrates critical value in complex reasoning tasks (Numeracy +3.0% and Position +6.3%), suggesting that even scaled-up models suffer from depth-wise feature degradation in challenging scenarios.

**Mitigating Prompt Forgetting.** Fig. 1 provides further evidence that Prompt Reinjection directly addresses depth-wise prompt forgetting. With Prompt Reinjection enabled, the layer-wise probing accuracy becomes largely stable and stays close to the shallow-layer level, rather than dropping with depth. This indicates that reinjection effectively preserves prompt-related signals throughout denoising, alleviating prompt forgetting in MMDiT.

**Visual Quality Preservation.** Crucially, the enhanced instruction adherence does not compromise image fidelity. As shown in Table 2, our method maintains or marginally improves performance across human-preference metrics (HPSv2, ImageReward, PickScore) and global alignment scores (CLIP). This indicates that our method effectively disentangles instruction adherence from generative quality, rectifying semantic drift without introducing artifacts.

**Qualitative Analysis.** We further provide qualitative comparisons across diverse instruction types in Fig. 5, with additional results in Appendix G. Under identical noise initialization, our method more consistently satisfies prompt constraints than the base models under varied prompt styles.

*Table 4.* Target-layer ablation on SD3-medium using GenEval overall score. We fix the origin layer to $l_{ori}{=}1$ and the reinjection weight to $w{=}0.025$. We vary the target-layer coverage by injecting into layers $l \in [\text{Start}, \text{End}]$ with a Stride.

| Strategy | Start | End | Stride | GenEval (Ovr.) |
|---|---|---|---|---|
| Full Layers | 2 | 23 | 1 | 0.7054 |
| Range Restriction | 2 | 11 | 1 | 0.6991 |
| | 12 | 23 | 1 | 0.6923 |
| Stride Sampling | 2 | 23 | 2 | 0.6947 |
| | 2 | 23 | 3 | 0.6959 |

*Table 5.* Ablation study on GenEval. ✓ and × denote the enabled and disabled statuses of LN and Rotation, respectively. The table reports the corresponding overall GenEval scores.

| Anchor | Rotation | SD3 | FLUX |
|---|---|---|---|
| × | × | 0.6849 | 0.6816 |
| ✓ | × | 0.6897 | 0.6823 |
| × | ✓ | 0.6910 | 0.6877 |
| ✓ | ✓ | 0.7054 | 0.7002 |

Overall, these quantitative and qualitative results suggest that reinjecting shallow-layer text features offers a simple and effective way to alleviate the prompt forgetting.

### 6.3. Ablation Studies and Discussions

We conduct a comprehensive ablation study on SD3-medium and FLUX.1-dev to assess the impact of key design choices: origin layer selection ($l_{ori}$), reinjection weight ($w$), and the constituent components of our alignment pipeline. We further evaluate the robustness of Prompt Reinjection to guidance scales and analyze its computational overhead.

**Origin Layer and Reinjection Weight.** We sweep the origin layer $l_{ori}$ (the layer from which text features are extracted) and the injection weight $w$ (Tables 10 and 11 in appendix). A consistent pattern emerges across architectures: a "shallow-source, low-weight" regime ($l_{ori} \in \{1, 2, 4\}, w < 0.1$) yields the most robust improvements in instruction following. This corroborates our probing findings (Sec. 4.2), which identify shallow layers as the richest reservoir of recoverable semantic information.

Notably, the optimal $l_{ori}$ correlates with the distributional characteristics observed in Sec. 4.1. SD3 favors $l_{ori} = 1$, while FLUX favors $l_{ori} = 2$. This aligns with the PCA projections (Fig. 2): the most effective origin layer is located immediately after the initial transition phase where text features adapt to the visual latent space. Selecting $l_{ori}$ at this stable inflection point minimizes distributional incompatibility while maximizing semantic fidelity.

**Target Layer Coverage.** We investigate the optimal depth range for injection in Table 4. Applying Prompt Reinjection to the full sequence of subsequent blocks ($L_{tgt} = \{l \mid l > l_{ori}\}$) consistently outperforms restricted strategies (injecting only into early or late blocks) and strided injection. This

*Table 6.* CFG ablation on SD3-medium using GenEval overall score. We fix the reinjection configuration to $l_{ori}{=}1$, $L_{tgt}{=}\{l \mid l > l_{ori}\}$, and $w{=}0.025$, and vary the classifier-free guidance scale.

| CFG Scale | 4.0 | 6.0 | 7.0 | 8.0 |
|---|---|---|---|---|
| SD3 (base) | 0.677 | 0.691 | 0.679 | 0.679 |
| SD3 + Ours | 0.696 | 0.702 | 0.706 | 0.701 |

*Table 7.* Compute and memory overhead on SD3-medium per target block. **PR** = Prompt Reinjection; **A** = distribution anchoring/restoration; **R** = orthogonal Procrustes geometry alignment. **Rel.** reports FLOPs as a fraction of one SD3 transformer block. Latency is measured per target block on H200 GPU; memory overhead is estimated for FP16/BF16 (2 bytes/element).

| Comp. | FLOPs | Rel. | Lat. (ms) | Mem. (MB) |
|---|---|---|---|---|
| SD3 block | $1.53 \times 10^{12}$ | 1.0000 | 2.118 | – |
| +PR (w/o A, w/o R) | $+3.28 \times 10^{7}$ | 1.0000 | 2.148 | +1.6 |
| +PR (A) | $+3.60 \times 10^{8}$ | 1.0002 | 2.261 | +6.3 |
| +PR (A+R) | $+1.35 \times 10^{11}$ | 1.0883 | 2.291 | +7.8 |

suggests that countering prompt forgetting requires continuous, dense semantic reinforcement throughout the deeper transformer layers ensure it remains effective throughout the deeper layers.

**Impact of Alignment Components.** Based on Tables 10 and 11, We fix each model to its best-performing ($l_{ori}, w$) and ablate the two alignment components. Table 5 shows that both distribution anchoring and restoration (Anchor) and geometry alignment (Rotation) contribute additional gains, with geometry alignment providing the larger improvement. Notably, even without either alignment component, simple prompt reinjection still outperforms the base model, indicating that the injection mechanism itself is effective; alignment primarily improves stability and unlocks stronger gains by reducing cross-layer mismatch.

**Robustness to CFG.** Table 6 shows that our gains are stable across a wide CFG range. Enabling reinjection improves GenEval at all tested CFG scales, and the relative improvement remains consistent, indicating that the method is not sensitive to the particular CFG choice under the default inference setup.

**Inference Cost.** Table 7 shows that *Prompt Reinjection* adds only a modest additional per-block overhead compared to a native SD3 transformer block. Rotation-based geometry alignment is the main contributor to extra compute and memory. Overall, *Prompt Reinjection* provides a favorable efficiency–effectiveness trade-off for improving instruction following at inference time.

## 7. Conclusion

We identify *prompt forgetting* in MMDiTs: prompt information in the text branch progressively degrades with depth during denoising, as evidenced by CKNNA/PCA analyses and layer-wise probes. To address this, we propose *Prompt Reinjection*, a training-free inference-time method

that reinjects aligned shallow text features into deeper blocks. *Prompt Reinjection* consistently improves instruction following while maintaining overall image quality.

## Impact Statement

Our method improves MMDiTs' ability to follow instructions about spatial layout, attribute binding, and style. While this can support legitimate creative applications, it may also lower the barrier to generating imitative or deceptive content, including deepfakes, identity impersonation, and copyrighted style imitation.

The goal of this work is to reduce semantic degradation at the architectural level, rather than to enable unethical forgery. In deployment, such techniques should be paired with safeguards such as watermarking, content fingerprinting, and safety filters.

## Acknowledgements

This project is sponsored by Natural Science Foundation of Shanghai under Grant No. 24ZR1407200, Shanghai Oriental Talents Project under Grant No. QNKJ2024060, and Shanghai Municipal Commission of Economy and Informatization (No. 2025-GZL-RGZN-BTBX-01011).

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

# A. Limitation and Future Work

**Limitation.** Although Prompt Reinjection incorporates alignment, it is still hindered by cross-depth discrepancies in feature distribution and geometry. As a result, we can not reliably use large reinjection weights, and the choice of origin layer is also constrained by cross-layer compatibility. This can affect stability and ease of use: for a new MMDiT, achieving optimal performance may require manual selection of the reinjection weight and origin layer.

**Future Work.** Future work can explore stronger alignment or more expressive reinjection designs to further improve cross-layer semantic transfer. Instead of using a shared reinjection weight, it is also promising to learn layer- or timestep-dependent $w$ for more precise control. In addition, fine-tuning models with *Prompt Reinjection* may reduce side effects and enable much stronger reinjection (e.g., $w > 0.1$). Finally, a more fundamental direction is to add direct supervision to the text branch during training (e.g., a text reconstruction loss) to encourage semantic preservation throughout the stack.

# B. Supplementary Details of Analytical Experiments

## B.1. CKNNA: Measuring Local Semantic-Structure Preservation

We adopt Conditional $k$-Nearest Neighbor Alignment (CKNNA) (Huh et al., 2024) to quantify how well the *local semantic structure* of a representation is preserved across two feature spaces. Here, *local semantic structure* refers to the *neighborhood relations* among tokens: tokens that are semantically similar (e.g., describing the same object, attribute, or relation) should remain close to each other under the representation's similarity metric. Preserving this neighborhood structure matters because it maintains fine-grained, token-level distinctions that downstream attention and composition mechanisms rely on; when neighborhoods collapse or reshuffle, prompt-related semantics become less separable and harder to recover, consistent with prompt forgetting.

Given two feature matrices $A, B \in \mathbb{R}^{N \times D}$ for the same set of $N$ tokens, we first compute pairwise similarities using a cosine kernel:

$$K_{ij}^A = \frac{\langle A_i, A_j \rangle}{|A_i|_2 \cdot |A_j| * 2 + \epsilon}, \quad K^B * ij = \frac{\langle B_i, B_j \rangle}{|B_i|_2 \cdot |B_i|_2 + \epsilon}. \tag{12}$$

To reduce global bias and make similarities comparable across spaces, we apply standard kernel centering:

$$\tilde{K}^A = HK^AH, \quad \tilde{K}^B = HK^BH, \quad H = I - \frac{1}{N}\mathbf{1}\mathbf{1}^\top, \tag{13}$$

where $\mathbf{1}$ is the all-ones vector.

For each token $i$, let $\mathcal{N}_k^A(i)$ denote the indices of the top-$k$ most similar tokens under $\tilde{K}^A$ (excluding $i$ itself), i.e., the $k$ largest off-diagonal entries in row $i$. We define $\mathcal{N}_k^B(i)$ analogously under $\tilde{K}^B$. CKNNA is then computed as the average overlap of the two $k$-NN sets:

$$\text{CKNNA} * k(A, B) = \frac{1}{N} \sum *i = 1^N \frac{|\mathcal{N}_k^A(i) \cap \mathcal{N}_k^B(i)|}{k}. \tag{14}$$

In our analysis, we set $A = T^{(0)}$ (text-encoder outputs, i.e., the input text-token space) and $B = T^{(l)}$ (the text-token features at MMDiT layer $l$). A lower $\text{CKNNA}_k$ indicates that tokens that were locally similar in $T^{(0)}$ are less likely to remain neighbors in $T^{(l)}$, implying that the representation increasingly disrupts token-level neighborhood relations and thus weakens the preservation of local semantic structure as depth increases.

## B.2. Layer-wise Probing: Token-Category Recoverability

We quantify text-feature degradation via a controlled probing experiment that measures *token-category recoverability* from intermediate text representations. The task is a token-category classification with five coarse categories: *noun*, *adjective*, *spatial-relation*, *numeral*, and *others*.

**Prompt Set and Labels.** We use GenEval prompts and construct 499 training prompts and 54 test prompts. After removing the fixed prefix "a photo of", we assign each remaining word one category label from the five classes above. When a word is segmented into multiple sub-tokens by the tokenizer, we propagate the word-level label to all corresponding sub-tokens to obtain token-level supervision.

**Layer-wise Feature Extraction.** For each prompt, we run a single forward denoising pass and extract text-token features from all MMDiT layers. We remove padding tokens and account for special tokens so that feature vectors and token labels remain aligned under a consistent token index order. This yields a matched set of (feature, label) pairs for every layer.

**Probes and Controlled Training.** We train one lightweight MLP probe per layer, using the *same* architecture and training configuration (optimizer, learning rate , batch size, and number of epochs). We use the Adam optimizer with a fixed learning rate of 1e-4, a batch size of 64, and train for 50 epochs. Each probe takes that layer's token features as input and predicts the token category. By keeping probe capacity and training protocol fixed, differences in performance across layers can be attributed to the information content of the representations rather than probe-specific confounds.

**Metric and Interpretation.** We evaluate each layer-specific probe on the held-out test prompts and report classification accuracy. Higher accuracy indicates higher token-category recoverability and thus stronger retention of token-level

*Table 8.* Model information and default settings during inference.

| Models | SD3-medium | SD3.5-large | FLUX.1-Dev | Qwen Image |
|---|---|---|---|---|
| MMDiT Blocks | [0, 23] | [0, 37] | [0, 57] | [0, 59] |
| Parameters | 2B | 8B | 12B | 20B |
| Inference Steps | 28 | 28 | 50 | 50 |
| CFG Scale | 7.0 | 3.5 | 3.5 | 4.0 |
| Size | (1024, 1024) | (1024, 1024) | (1024, 1024) | (1024, 1024) |
| Origin Layer | 1 | 2 | 2 | 30 |
| Target Layers | 2-23 | 3-37 | 3-57 | 31-59 |
| Injection Weight | 0.025 | 0.025 | 0.025 | 0.025 |

*Table 9.* Calibration-dataset ablation for Procrustes alignment on SD3-medium using GenEval overall score. We fix $l_{ori}=1$, $L_{tgt}=\{l \mid l > l_{ori}\}$, and $w=0.025$, and vary the prompt set used to collect text-token pairs for computing the orthogonal mapping. Abbreviations: C5K = COCO-5K (Lin et al., 2014); B1K/B5K/B10K = BLIP3o subsets (Chen et al., 2025) with 1K/5K/10K prompts; E5K = Echo-4o-Image subset (Ye et al., 2025) with 5K prompts.

| Calib. set | C5K | B1K | B5K | B10K | E5K |
|---|---|---|---|---|---|
| GenEval (Ovr.) | 0.706 | 0.696 | 0.699 | 0.695 | 0.697 |

linguistic signals in that layer. A depth-wise decrease in accuracy implies that token-level prompt information becomes less recoverable from deeper text representations.

## C. Detailed Setup of Evaluation Results

All quantitative and qualitative comparisons reported in the main paper (excluding ablations) follow the model information and default inference settings summarized in Table 8. Specifically, for each model (SD3-medium, SD3.5-large, FLUX.1-dev, and Qwen-Image), we use its official default sampling configuration (number of inference steps, CFG scale, and $1024\times1024$ resolution), and keep these inference settings identical between the base model and the base model with Prompt Reinjection enabled.

These Prompt Reinjection settings are chosen based on the best-performing combinations identified in our ablation studies. For SD3-medium, SD3.5-large, and FLUX.1-dev, the chosen origin layer is a shallow block after the initial sharp feature transition, and reinjection is applied to all subsequent blocks to sustain prompt throughout the deeper stack.

For Qwen-Image, we observe that using a mid-stack origin layer yields a more noticeable improvement than using very shallow layers, and thus set the origin and target layers accordingly (Table 8). Due to the larger model size and higher inference cost, we do not perform a search as exhaustive as SD3 and FLUX.

## D. Detailed Analysis of Prompt Reinjection

**Calibration Data Choice.** Table 9 shows that COCO-5K prompts yield the best GenEval score after alignment. We attribute this to the broad and diverse nature of COCO captions (Lin et al., 2014), which cover richer token distributions for estimating a stable cross-layer orthogonal mapping.

*Table 10.* Ablation study for SD3-medium (baseline = 0.6793). We evaluate combinations of origin layer and injection weight $w$, reporting the overall GenEval score. Red cells indicate scores higher than the baseline; dark red cells highlight the optimal results.

| Origin layer | Injection Weight $w$ | | | |
|---|---|---|---|---|
| | 0.01 | 0.025 | 0.05 | 0.1 |
| 0 | 0.6901 | 0.6853 | 0.6593 | 0.2433 |
| 1 | 0.6946 | 0.7054 | 0.6967 | 0.5858 |
| 2 | 0.6919 | 0.7010 | 0.6814 | 0.5576 |
| 4 | 0.6923 | 0.6871 | 0.6998 | 0.5915 |
| 6 | 0.6871 | 0.6831 | 0.6859 | 0.6270 |
| 8 | 0.6873 | 0.6831 | 0.6817 | 0.6404 |
| 12 | 0.6811 | 0.6776 | 0.6686 | 0.6552 |
| 16 | 0.6883 | 0.6827 | 0.6792 | 0.6803 |

*Table 11.* Ablation study for FLUX.1-dev (baseline = 0.6613). We evaluate combinations of origin layer and injection weight $w$, reporting the overall GenEval score. Red cells indicate scores higher than the baseline; dark red cells highlight the optimal results.

| Origin layer | Injection Weight $w$ | | | |
|---|---|---|---|---|
| | 0.01 | 0.025 | 0.05 | 0.1 |
| 0 | 0.6879 | 0.6865 | 0.6129 | 0.4863 |
| 1 | 0.6905 | 0.6938 | 0.6515 | 0.5596 |
| 2 | 0.6864 | 0.7002 | 0.6756 | 0.6010 |
| 4 | 0.6985 | 0.6986 | 0.6775 | 0.6187 |
| 8 | 0.6845 | 0.6910 | 0.6821 | 0.6428 |
| 16 | 0.6449 | 0.6329 | 0.5859 | 0.5088 |
| 24 | 0.6634 | 0.6555 | 0.6342 | 06050 |
| 32 | 0.6586 | 0.6600 | 0.6545 | 0.6583 |

Notably, using alternative prompt collections—different-sized subsets of BLIP3o (Chen et al., 2025) or Echo-4o-Image (Ye et al., 2025)—produces very similar results, indicating that Procrustes calibration is fairly robust to the specific prompt source as long as the dataset is diverse.

## E. Additional Results on HunyuanImage-2.1

We conduct additional experiments on HunyuanImage-2.1, a 17B-parameter model. We use the same setting of origin = 1 and set $w = 0.025$, without exhaustive hyperparameter tuning. The quantitative results on GenEval and DPG-Bench are reported in Table 12 and Table 13, respectively.

## F. Comparison with Other MMDiT-focusing Method

We compare against TACA (Lv et al., 2025) because it is a recent method that explicitly studies cross-modal interaction in MMDiT-based text-to-image models and improves instruction following by strengthening textual conditioning during denoising. Unlike our training-free Prompt Reinjection, TACA requires LoRA fine-tuning of the model.

Table 14 compares FLUX with TACA (LoRA rank r=64) and our Prompt Reinjection on GenEval. TACA yields

*Table 12.* **Quantitative** comparison on **GenEval** for **HunyuanImage-2.1 (17B)**. We compare the base model with **ours** under origin=1 and $w$=0.025.

| Model | GenEval | | | | | | |
|---|---|---|---|---|---|---|---|
| | Overall | Single obj. | Two obj. | Counting | Colors | Color attr | Position |
| HunyuanImage | 0.7708 | 0.9906 | 0.9116 | 0.5906 | 0.8697 | 0.6475 | 0.6150 |
| HunyuanImage + Ours | 0.8305 | 1.0000 | 0.9394 | 0.7438 | 0.9149 | 0.6775 | 0.7075 |

*Table 13.* **Quantitative** comparison on **DPG-Bench** for **HunyuanImage-2.1 (17B)**. We compare the base model with **ours** under origin=1 and $w$=0.025.

| Model | DPG-Bench | | | | | |
|---|---|---|---|---|---|---|
| | Overall | Other | Attribute | Entity | Global | Relation |
| HunyuanImage | 85.4419 | 82.4000 | 89.2409 | 91.8901 | 81.7629 | 94.3520 |
| HunyuanImage + Ours | 86.3294 | 84.0000 | 89.7390 | 92.4233 | 82.9787 | 94.1973 |

its largest gain on the Position subtask, while our method achieves a higher overall score and improves most categories. This suggests that directly reintroducing high-fidelity shallow text features provides a broader boost across diverse constraint types.

## G. Qualitative Comparisons

This appendix presents the results of our qualitative comparison experiments conducted on SD3, SD3.5, FLUX, and Qwen-Image. Evaluated on a diverse set of text prompts, our method achieves superior text–image consistency relative to base models across key dimensions.

## H. Supplementary CKNNA and PCA Results

This appendix reports the layer-wise CKNNA and PCA projection results for SD3, SD3.5, and FLUX, which serve as supplementary materials to Figure 2 in the main text.

## I. Limitations of Probing and Higher-Capacity Probe Results

We acknowledge that declining probing accuracy does not by itself prove literal information loss in a strict information-theoretic sense. Probing performance may also decrease when information is reorganized into more distributed or compositional representations that are harder to decode. Therefore, our probing results should be interpreted as evidence of reduced recoverability of token-level instruction information, rather than as direct proof that such information is completely removed.

To examine whether the observed degradation is mainly caused by such representational reorganization, we conduct additional experiments with higher-capacity probes, including 3-layer and 5-layer MLP probes, as well as sequence-level Transformer decoders with 2, 3, and 4 Transformer blocks. The results are reported in Table 15. Rows correspond to MMDiT layer indices, and columns correspond to denoising timesteps along the reverse diffusion process.

Across all probe families, increasing probe capacity brings

*Table 14.* **Quantitative** comparison on **GenEval** for **FLUX.1-dev** under three inference-time variants: the base model, **TACA** (LoRA rank $r$=64), and **ours** (Prompt Reinjection). All results use the official default inference settings for each variant, and Prompt Reinjection configuration matches the main comparisons (C).

| Model | GenEval | | | | | | |
|---|---|---|---|---|---|---|---|
| | Overall | Single obj. | Two obj. | Counting | Colors | Color attr | Position |
| FLUX | 0.6613 | 0.9875 | 0.8308 | 0.7281 | 0.7686 | 0.4575 | 0.1950 |
| FLUX + TACA | 0.6793 | 0.9969 | 0.8384 | 0.7312 | 0.7766 | 0.4600 | 0.2723 |
| FLUX + Ours | 0.6986 | 0.9969 | 0.8485 | 0.7500 | 0.8165 | 0.5275 | 0.2525 |

only marginal gains, while the decline in deeper MMDiT layers remains consistent. This suggests that stronger probes are still unable to substantially recover token-level instruction information from deep-layer representations, providing limited support for the hypothesis that the degradation is merely benign representational reorganization.

We therefore revise our interpretation more carefully: the probing results do not prove information loss in a strict sense, but they provide strong empirical evidence that token-level instruction information becomes progressively less recoverable and less usable in deeper layers. This interpretation is further supported by the alignment between declining probing accuracy and degraded instruction-following behavior, as well as the fact that prompt reinjection partially restores both probing performance and generation quality.

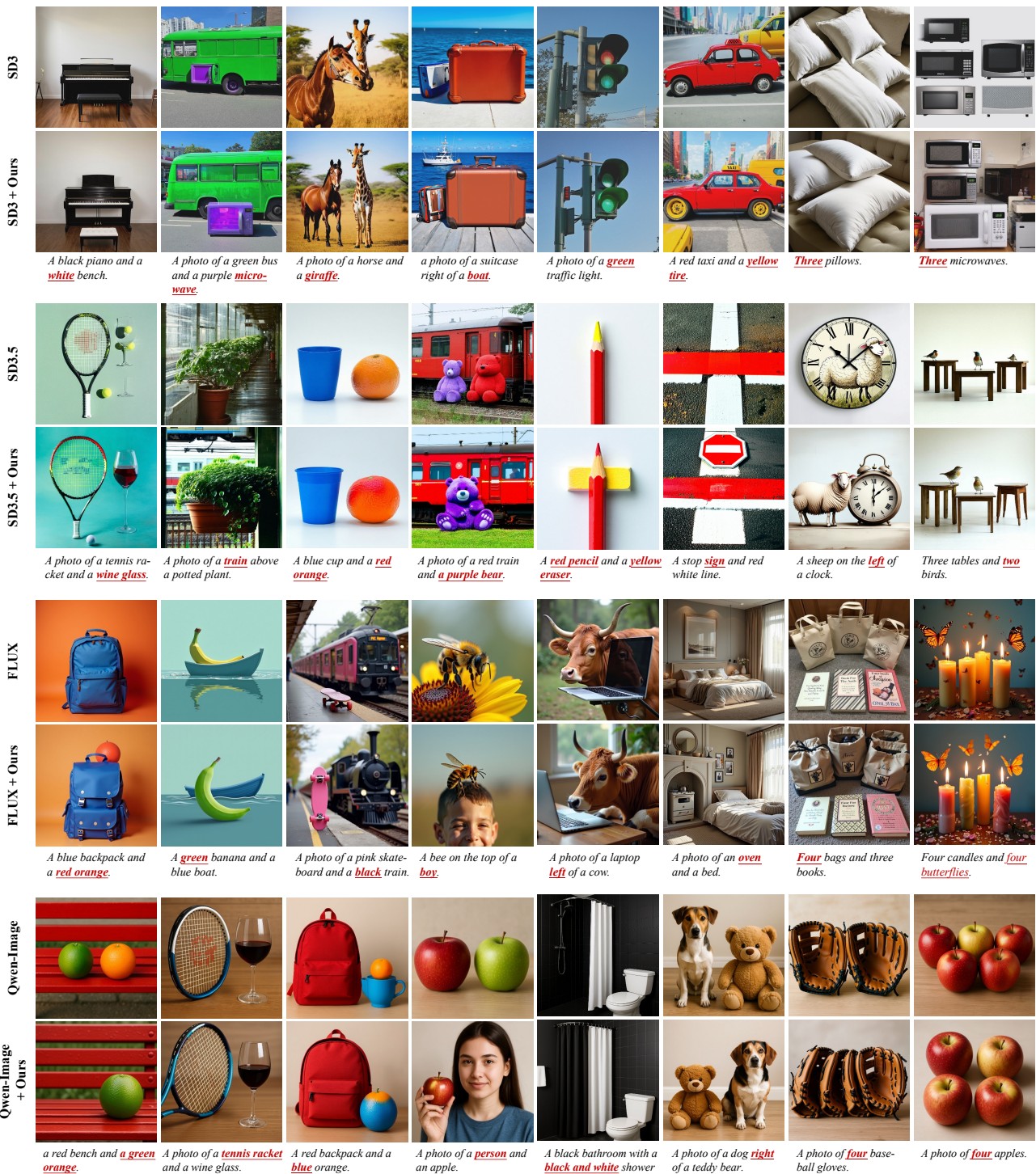

*Figure 6.* **Qualitative** between base models (SD3-medium, SD3.5-large, FLUX.1-Dev, and Qwen-Image) and their counterparts with our method enabled. The bold text in the prompts highlights specific constraints where our method significantly improves text-image consistency compared to the baselines.

| w/o. Ours | w/. Ours | w/o. Ours | w/. Ours |
|:---:|:---:|:---:|:---:|

**SD3**

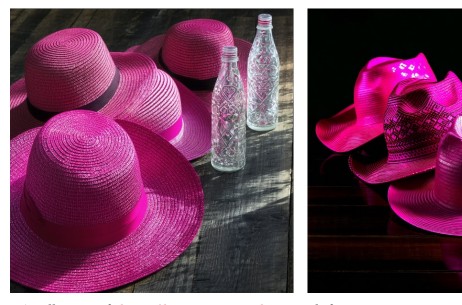 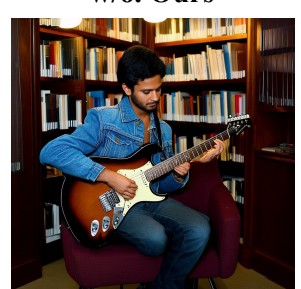

*A tall, gray tower looms over the bustling street below, where cars and buses navigate through the flow of traffic. The street is canopied by a row of leafy green trees, which cast dappled shadows onto the asphalt. Behind a ruddy __red car__ parked along the side of the road, more trees with thick foliage provide a backdrop of natural green against the urban environment. A large __yellow bus__ makes its way down the lane, adding vibrancy to the cityscape*

*A focused individual with a blue denim jacket is strumming an electric guitar amidst the quietude of a library. Surrounded by towering wooden bookshelves filled with an array of books, he is seated on a simple chair with a burgundy cushion. His guitar, a sleek __black instrument__ with silvery strings, catches the light from the overhead lamps as he creates a melody in this uncommon setting.*

**SD3.5**

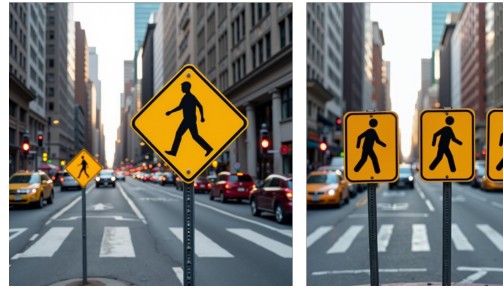 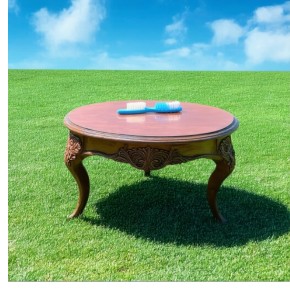 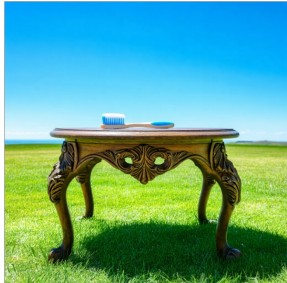

*A collection of __three vibrant magenta hats__, each featuring a unique pattern and texture, are arranged side by side on a dark, polished wooden surface. Nearby, two translucent bottles with intricate designs reflect the ambient light. The bottles are carefully positioned to the right of the hats, and their contents cast a slight shadow on the wood grain.*

*In an open outdoor setting, a decorative coffee table with elaborate wood carvings and curved legs is positioned under the expansive blue sky of a clear afternoon. On the surface of the table, an out-of-place toothbrush with __white and blue__ bristles sits alone. The table stands on a patch of vibrant green grass, and no other items or furniture are immediately visible in the vicinity.*

**FLUX**

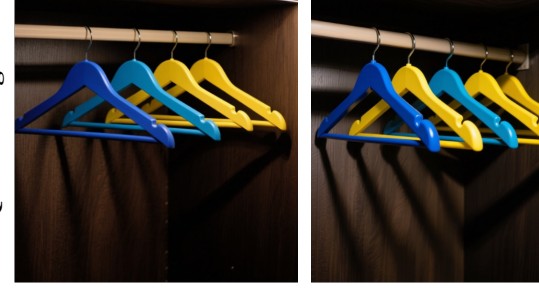 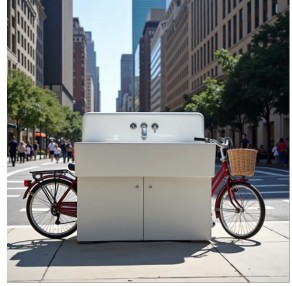 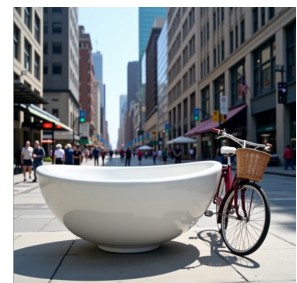

*At a busy city crossroads, __three__ square-shaped signs, featuring a bold crosswalk symbol in bright yellow, command attention from pedestrians and motorists alike. Positioned against the backdrop of a bustling urbanscape, the signs possess a reflective quality, enhancing their visibility even amidst the chaotic street movement. Anchored securely to the pavement, these uniform signs present the familiar pedestrian crossing imagery, delineating safe walking zones in an area dense with traffic.*

*In the bustling heart of the city, under the clear midday sky, stands a sparkling white sink, its porcelain surface gleaming in the sunlight. It is an unusual sight: the sink, vastly larger than a nearby old burgundy __bicycle__ with a wicker basket, positioned as if it were a modern art installation. The surrounding concrete ground of the cityscape contrasts sharply with the clean and polished texture of the oversized sink.*

**Qwen-Image**

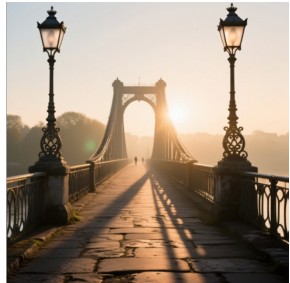 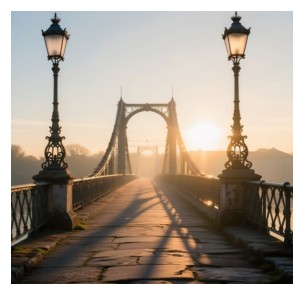

*In the dimly lit interior of a spacious wardrobe, a neat row of __five hangers__ stands out, each one a different brilliant hue—ranging from a deep royal blue to a bright lemon yellow. They are spaced evenly apart, casting soft shadows against the dark wooden back of the closet. The smooth, plastic texture of the hangers contrasts with the rough texture of the wardrobe's interior, and their curved shapes seem almost to beckon items of clothing to drape over them.*

*A picturesque bridge bathed in the warm glow of the early morning sun, flanked by two tall antique street lights. The street lights, with their ornate metalwork and frosted glass, cast elongated shadows across the weathered stone pathway of the bridge. The tranquil scene is further accentuated by the __absence of pedestrians__, giving the impression of a moment frozen in time just after dawn.*

*Figure 7.* **Qualitative** comparison between base models (SD3-medium, SD3.5-large, FLUX.1-Dev, and Qwen-Image) and their counterparts with our method enabled on complex prompts. The bold text in the prompts highlights specific constraints where our method significantly improves text-image consistency compared to the baselines.

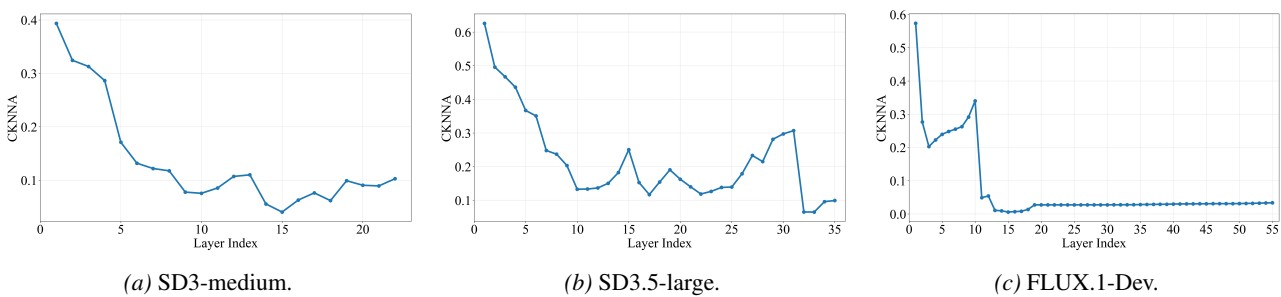

*(a)* SD3-medium.        *(b)* SD3.5-large.        *(c)* FLUX.1-Dev.

*Figure 8.* Layer-wise CKNNA analysis across SD3-medium, SD3.5-large, and FLUX.1-Dev.

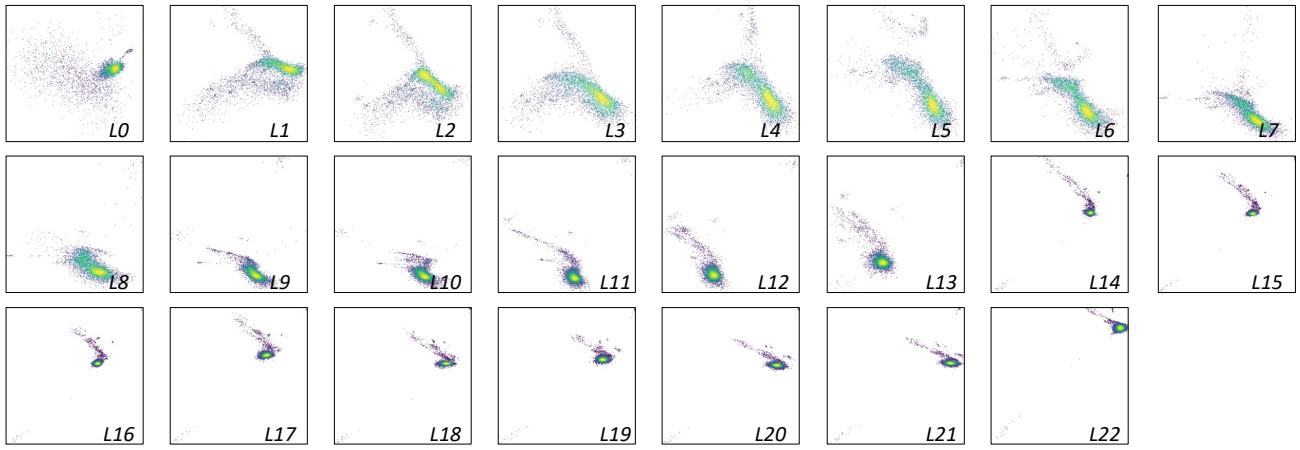

*Figure 9.* PCA visualization of intermediate text features for SD3-medium.

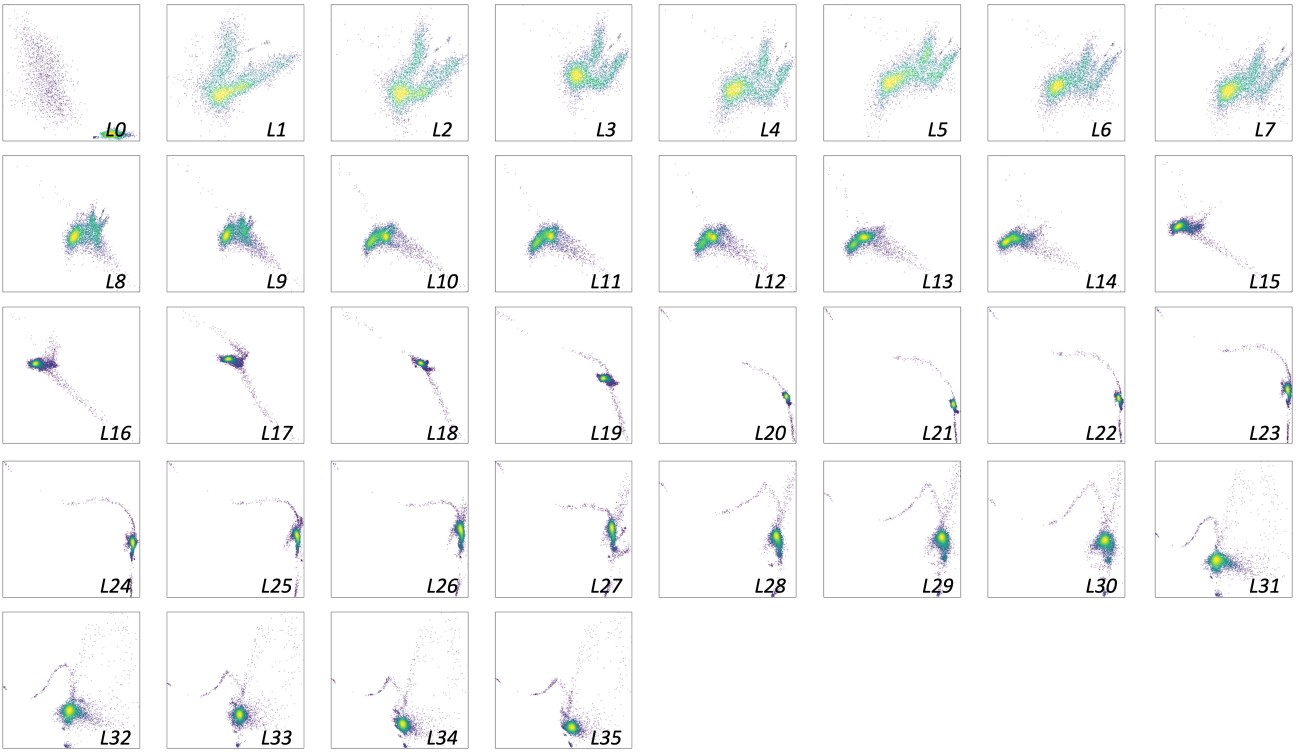

*Figure 10.* PCA visualization of intermediate text features for SD3.5-large.

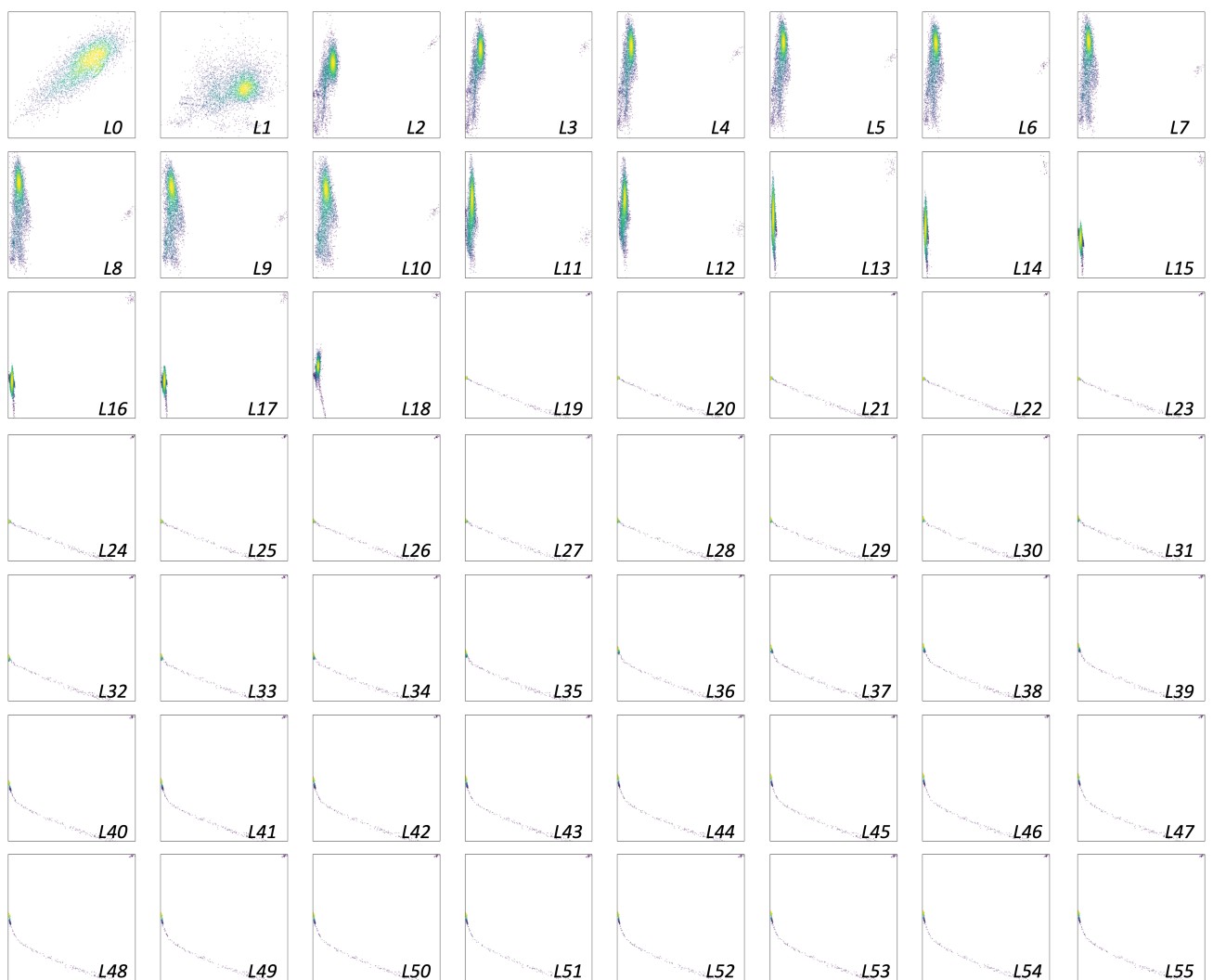

*Figure 11.* PCA visualization of intermediate text features for FLUX.1-dev.

*Table 15.* Higher-capacity probing results across MMDiT layers and denoising timesteps. Rows denote MMDiT layer indices, and columns denote denoising timesteps along the reverse diffusion process.

**1-layer MLP**

| Layer | $t=900$ | $t=700$ | $t=500$ | $t=300$ | $t=100$ |
|---|---|---|---|---|---|
| 0 | 0.9922 | 0.9961 | 0.9961 | 0.9961 | 0.9942 |
| 9 | 0.9922 | 0.9883 | 0.9903 | 0.9922 | 0.9922 |
| 18 | 0.9942 | 0.9942 | 0.9864 | 0.9942 | 0.9844 |
| 24 | 0.9805 | 0.9903 | 0.9864 | 0.9844 | 0.9747 |
| 27 | 0.9767 | 0.9747 | 0.9650 | 0.9533 | 0.9514 |
| 30 | 0.9494 | 0.9533 | 0.9475 | 0.9319 | 0.8930 |
| 33 | 0.9358 | 0.9630 | 0.9397 | 0.8988 | 0.9047 |
| 36 | 0.9300 | 0.9475 | 0.9377 | 0.9125 | 0.8813 |

**3-layer MLP**

| Layer | $t=900$ | $t=700$ | $t=500$ | $t=300$ | $t=100$ |
|---|---|---|---|---|---|
| 0 | 0.9961 | 0.9864 | 0.9942 | 0.9922 | 0.9922 |
| 9 | 0.9903 | 0.9942 | 0.9883 | 0.9922 | 0.9922 |
| 18 | 0.9942 | 0.9961 | 0.9961 | 0.9922 | 0.9922 |
| 24 | 0.9805 | 0.9903 | 0.9864 | 0.9844 | 0.9689 |
| 27 | 0.9650 | 0.9747 | 0.9611 | 0.9553 | 0.9358 |
| 30 | 0.9630 | 0.9669 | 0.9397 | 0.9397 | 0.9027 |
| 33 | 0.9339 | 0.9572 | 0.9455 | 0.9222 | 0.8949 |
| 36 | 0.9455 | 0.9591 | 0.9416 | 0.9163 | 0.8735 |

**5-layer MLP**

| Layer | $t=900$ | $t=700$ | $t=500$ | $t=300$ | $t=100$ |
|---|---|---|---|---|---|
| 0 | 0.9922 | 0.9903 | 0.9922 | 0.9903 | 0.9942 |
| 9 | 0.9942 | 0.9942 | 0.9942 | 0.9942 | 0.9942 |
| 18 | 0.9922 | 0.9883 | 0.9922 | 0.9922 | 0.9903 |
| 24 | 0.9844 | 0.9864 | 0.9864 | 0.9805 | 0.9514 |
| 27 | 0.9767 | 0.9669 | 0.9708 | 0.9358 | 0.9280 |
| 30 | 0.9494 | 0.9611 | 0.9533 | 0.9241 | 0.9066 |
| 33 | 0.9416 | 0.9591 | 0.9358 | 0.9027 | 0.8949 |
| 36 | 0.9416 | 0.9455 | 0.9319 | 0.9008 | 0.8716 |

**2 Transformer blocks**

| Layer | $t=900$ | $t=700$ | $t=500$ | $t=300$ | $t=100$ |
|---|---|---|---|---|---|
| 0 | 0.9864 | 0.9864 | 0.9903 | 0.9922 | 0.9903 |
| 9 | 0.9922 | 0.9922 | 0.9942 | 0.9883 | 0.9922 |
| 18 | 0.9922 | 0.9942 | 0.9961 | 0.9903 | 0.9883 |
| 24 | 0.9805 | 0.9864 | 0.9883 | 0.9669 | 0.9591 |
| 27 | 0.9844 | 0.9767 | 0.9786 | 0.9611 | 0.9436 |
| 30 | 0.9514 | 0.9669 | 0.9611 | 0.9319 | 0.9222 |
| 33 | 0.9436 | 0.9650 | 0.9630 | 0.9358 | 0.9241 |
| 36 | 0.9377 | 0.9514 | 0.9630 | 0.9319 | 0.9241 |

**3 Transformer blocks**

| Layer | $t=900$ | $t=700$ | $t=500$ | $t=300$ | $t=100$ |
|---|---|---|---|---|---|
| 0 | 0.9883 | 0.9883 | 0.9942 | 0.9942 | 0.9922 |
| 9 | 0.9942 | 0.9942 | 0.9942 | 0.9864 | 0.9883 |
| 18 | 0.9942 | 0.9922 | 0.9961 | 0.9942 | 0.9942 |
| 24 | 0.9864 | 0.9922 | 0.9864 | 0.9825 | 0.9805 |
| 27 | 0.9825 | 0.9747 | 0.9708 | 0.9669 | 0.9475 |
| 30 | 0.9455 | 0.9533 | 0.9514 | 0.9377 | 0.9358 |
| 33 | 0.9319 | 0.9591 | 0.9591 | 0.9339 | 0.9066 |
| 36 | 0.9416 | 0.9630 | 0.9455 | 0.9319 | 0.8930 |

**4 Transformer blocks**

| Layer | $t=900$ | $t=700$ | $t=500$ | $t=300$ | $t=100$ |
|---|---|---|---|---|---|
| 0 | 0.9825 | 0.9903 | 0.9922 | 0.9922 | 0.9903 |
| 9 | 0.9942 | 0.9864 | 0.9942 | 0.9903 | 0.9883 |
| 18 | 0.9942 | 0.9922 | 0.9961 | 0.9903 | 0.9922 |
| 24 | 0.9805 | 0.9864 | 0.9494 | 0.9650 | 0.9728 |
| 27 | 0.9825 | 0.9708 | 0.9591 | 0.9455 | 0.9455 |
| 30 | 0.9553 | 0.9611 | 0.9630 | 0.9319 | 0.9300 |
| 33 | 0.9436 | 0.9630 | 0.9591 | 0.9300 | 0.9261 |
| 36 | 0.9339 | 0.9572 | 0.9514 | 0.9222 | 0.9027 |

