# OpenReview forum: "Prompt Reinjection: Alleviating Prompt Forgetting in Multimodal Diffusion Transformers"
_ICML.cc/2026/Conference — ICML 2026 regular_

### Official Review · Reviewer_Za9G · 2026-02-26

**Soundness:** 2
**Presentation:** 3
**Significance:** 2
**Originality:** 2
**Overall Recommendation:** 3
**Confidence:** 4

**Summary:**

This paper identifies a "prompt forgetting" problem in Multimodal Diffusion Transformers (MMDiTs), where fine-grained textual semantics in the text branch progressively degrade as network depth increases.
The authors attribute it to a supervisory asymmetry: the denoising loss operates only on visual latents, so text features receive no direct semantic supervision and are free to drift. They empirically characterize this effect across SD3, SD3.5, FLUX, and Qwen-Image using CKNNA analysis, PCA visualization, and layer-wise probing classifiers that decode token-level linguistic attributes from intermediate representations.
Building on these findings, the authors propose Prompt Reinjection, a training-free inference-time method that reinjects aligned shallow-layer text features into deeper transformer blocks. To handle cross-layer distributional and geometric mismatches, the method applies layer normalization for distribution anchoring and an orthogonal Procrustes transform for geometry alignment. Experiments on GenEval, DPG-Bench, and T2I-CompBench++ show consistent improvements in instruction following, particularly on spatial reasoning and attribute binding tasks, while maintaining image quality as measured by HPSv2, ImageReward, PickScore, and CLIP score.

**Compliance With Llm Reviewing Policy:**

Affirmed.

**Final Justification:**

The author's response has partially addressed my concerns. However, some residual issues remain, leading me to believe that there is still room for further improvement in the quality of this manuscript. Consequently, I am maintaining my negative rating while lowering my confidence level.

**Key Questions For Authors:**

**Q1.** Declining probing accuracy may reflect information reorganization into distributed or compositional codes rather than true information loss, which is a well-known limitation of probing methodology, as discussed by Hewitt & Liang (2019) [1].
Have you tried higher-capacity probes such as deeper MLPs or sequence-level decoders to rule this out? If stronger probes also fail to recover the attributes, my soundness score would increase.

**Q2.** Probing is conducted at a single unspecified denoising timestep. Does the degradation pattern hold uniformly across timesteps, or is it concentrated at specific noise levels? Is the reinjection benefit also timestep-dependent?

**Q3.** Qwen-Image requires origin layer 30, a mid-stack position, which contradicts the premise that shallow layers retain the most semantics. Why do shallow layers fail here? What principled guidance exists for selecting the origin layer on a new model without exhaustive search?

**Q4.** Results lack error bars. Some gains are small, such as FLUX on DPG improving by only 0.71 points, and a few metrics actually regress, notably FLUX PickScore and Qwen-Image CLIP. Are results averaged over multiple seeds? Are the regressions within noise or indicative of a systematic trade-off?


-----
**Ref**
[1] Hewitt, J., & Liang, P. (2019). Designing and Interpreting Probes with Control Tasks. Proceedings of the 2019 Conference on Empirical Methods in Natural Language Processing (EMNLP).

**Limitations:**

The authors provide a reasonable discussion in Appendix A, acknowledging sensitivity to reinjection weight and origin layer selection. However, there are still a few issues that need to be addressed.

First, failure mode analysis is missing. Tables 10 and 11 show severe degradation at w=0.1, but no qualitative analysis or explanation is offered for why large weights cause such harm.

Second, probing methodology limitations are unacknowledged. The known gap between probing accuracy and true information content is a core methodological concern and should be explicitly discussed rather than treated as settled. Related discussions can be found in Hewitt & Liang (2019) [1] and Voita & Titov (2020) [2].

-----
**Ref**
[1] Hewitt, J., & Liang, P. (2019). Designing and Interpreting Probes with Control Tasks. Proceedings of the 2019 Conference on Empirical Methods in Natural Language Processing (EMNLP).
[2] Voita, E., & Titov, I. (2020). Information-Theoretic Probing with Minimum Description Length. Proceedings of the 2020 Conference on Empirical Methods in Natural Language Processing (EMNLP).

**Strengths And Weaknesses:**

**Strengths**

S1. Systematic analysis.
The paper combines three complementary perspectives (CKNNA, PCA, layer-wise probes) to characterize text feature degradation across four MMDiT architectures spanning 2B to 20B parameters.

S2. Simple, practical, and training-free method.
Prompt Reinjection requires no parameter updates, only a lightweight one-time Procrustes calibration. This gives it strong practical appeal, similar in spirit to prior training-free interventions [1] but targeting a mechanism unique to MMDiTs.


**Weaknesses**

W1. Probing failure does not equal information loss.
Declining probe accuracy may reflect information reorganization into distributed codes rather than true forgetting, which is a limitation of probing methods [2, 3, 4].
The paper does not attempt higher-capacity probes or information-theoretic alternatives to rule out the reorganization hypothesis, which undermines the core theoretical claim.

W2. Qwen-Image contradicts the hypothesis.
Qwen-Image requires origin layer 30 (mid-stack), contradicting the claim that shallow layers retain the most semantics. No probing analysis is provided for Qwen-Image to explain this, raising concerns about generalizability without per-model tuning.

W3. No statistical rigor.
Results lack error bars. Some gains are small (FLUX DPG: +0.71) and some metrics regress (FLUX PickScore, Qwen-Image CLIP) without acknowledgment.

W4. Insufficient novelty differentiation.
Li et al. (2026), cited by the authors, studies layer-wise text feature roles and amplifies selected-layer features, which is highly similar. No experimental comparison is provided. The individual components (residual injection, normalization, Procrustes alignment) are all standard techniques, and the paper does not clearly articulate what makes their combination novel.

------
**References**

[1] Chefer, H., et al. (2023). Attend-and-Excite: Attention-Based Semantic Guidance for Text-to-Image Diffusion Models. *ACM TOG (SIGGRAPH)*, 42(4), 1–10. \
[2] Hewitt, J., & Liang, P. (2019). Designing and Interpreting Probes with Control Tasks. *EMNLP 2019*. \
[3] Voita, E., & Titov, I. (2020). Information-Theoretic Probing with Minimum Description Length. *EMNLP 2020*. \
[4] Belinkov, Y. (2022). Probing Classifiers: Promises, Shortcomings, and Advances. *Computational Linguistics*, 48(1), 207–219. \
[5] Balaji, Y., et al. (2022). eDiff-I: Text-to-Image Diffusion Models with an Ensemble of Expert Denoisers. *arXiv:2211.01324*.

---

> ### Author Rebuttal · Authors · 2026-03-31
>
> > **Response to Weakness 1, Question 1 and Limitation 2**
>
> Acknowledging the reviewer's point that probing failure $\neq$ info-loss, we employed high-capacity probes(3/5-layer MLPs, 2/3/4-block Transformers). These probes yielded only marginal gains (results in the anonymous link due to space constraints: https://anonymous.4open.science/r/probing_results-DD6B), suggesting information loss rather than a mere reorganization of codes. Practically, if even high-capacity probes fail to recover it, the model’s native attention mechanism is also unlikely to access it effectively for generation, which is further supported by the correlation between probing degradation and reduced instruction-following performance.
>
> ------
>
> > **Response to Question 2**
>
> Thank you for this insightful question. Our probing results (in **Question 1**) show that the degradation persists throughout the denoising trajectory and becomes more pronounced in deep layers at later timesteps. A likely reason is that once the main image content and attributes have been largely established, generation becomes less dependent on text conditioning, making text features more prone to drift.
>
> To identify the most effective reinjection timing, we tested three stages: S1 (1000–667), S2 (666–334), and S3 (333–0). The results are as follows:
>
> | Stage      | GenEval |
> | ---------- | ------- |
> | SD3-medium | 0.6793  |
> | All Stages | 0.7059  |
> | Stage 1    | 0.7018  |
> | Stage 2    | 0.6810  |
> | Stage 3    | 0.6796  |
>
> **The largest gain comes from Stage 1**, while later-stage reinjection is much less effective. This is likely because the main image content is largely determined early in the denoising process, making early semantic preservation more important.
>
> ---
>
> > **Response to Weakness 2 and Question 3**
>
> Thank you for this important comment. Qwen-Image is indeed a special case, and its best origin layer at 30 needs clarification.
>
> **For Qwen-Image, we observe unusual early-layer feature dynamics.** Before layer 30, feature magnitudes rise rapidly from about 0–1 to the fp16 limit and then stay near saturation due to explicit clipping. **We did not observe such severe numerical instability in the other models.** Accordingly, before layer 30, **its features are unstable and do not show the same progressive clustering pattern as other backbones**, whereas from layer 30 to 60 they exhibit a more similar trend. We therefore apply our method in the 30–60 range for it. The gain is smaller but still positive, suggesting that prompt forgetting and prompt reinjection still apply in this challenging case.
>
> For a new model, our default is simple: start from **early layers with a small weight** (e.g., origin = 1, w = 0.025), and only move deeper if the early layers are clearly abnormal, as in Qwen-Image. We verify this on **HunyuanImage-2.1 (17B)**, which is architecturally comparable to Qwen-Image (**both use a 60-layer MMDiT and Qwen2.5-VL 7B as the text encoder**). Without tuning, the default setting already improves GenEval from 0.77 to 0.83 and DPG-Bench from 85.44 to 86.33 (see response to **Reviewer fmRR, Weakness 2**).
>
> This suggests that **Qwen-Image is a special case caused by severe numerical instability**, not evidence against the general selection principle. We will revise the paper to make this guidance explicit.
>
> ---
>
> > **Response to Weakness 3 and Question 4**
>
> Thank you for raising this important point. We repeated the experiments (10 times) and now report **95% confidence intervals**. Due to space constraints, the full table is provided in the anonymous link:
>
> https://anonymous.4open.science/r/quantitative_results_95ci-6C96
>
> Overall, the results show **stable gains in instruction following** without **consist and systematic degradation in visual quality**. We will include the confidence intervals and discussion in the revised paper.
>
> ---
>
> > **Response to Weakness 4**
>
> Regarding **Li et al. (2026)**, their work focuses on the layer-wise roles of text features and enhances generation by scaling text features from selected layers using layer-specific scaling factors.
>
> In contrast, our work uniquely identifies and quantifies “Prompt Forgetting” in MMDiT architecture, proposing the targeted Prompt Reinjection mechanism. This contribution not only provides a practical solution but also highlights a previously overlooked architectural problem.
>
> ---
>
> > **Response to Limitation 1**
>
> For large injection weights (**w ≥ 0.1**), the main failures are **structural collapse** and **visual artifacts**. We believe this is because overly strong reinjection disrupts the normal denoising trajectory.
>
> Our two alignment modules can only **mitigate**, not fully remove the **distributional** and **geometric** mismatch between origin- and target-layer features. When w is large, the mismatch is amplified with the injected signal, destabilizing the target feature space and degrading image quality.

---

> > ### Author Rebuttal · Reviewer_Za9G · 2026-03-31
> >
> > The authors have provided detailed experiments and responses. However, the reviewer still has some concerns.
> > ---
> > **For W1/Q1**: Since the authors did not provide details regarding the higher-capacity methods, it remains difficult for the reviewer to assess the credibility of this technique. It's recommended that including these details in the revised version will be beneficial.
> >
> > **For W2/Q3**: Qwen-Image serves as the primary experimental model in this paper. So even with the addition of experiments involving Hunyuan, it remains difficult to attribute the observed issues solely to numerical instability as a specific anomaly unique to Qwen.
> >
> > **Regarding Q4**: The reviewers' concerns, specifically regarding the potential causes for the decline in performance metrics, have not yet been adequately addressed.
> >
> > ---
> > **Despite these issues**, the reviewer is willing to raise the score given the authors' detailed experiments since these questions can be addressed in the revision.

---

> > > ### Author Response · Authors · 2026-04-04
> > >
> > > > **Response to W1&Q1**
> > >
> > > We thank the reviewer for the constructive comments and provide the detailed probing setup below.
> > >
> > > **We split the Geneval dataset into training and test sets (9:1) with a roughly balanced distribution across prompt categories**, and manually annotate words into **five categories: numeral, locative word, noun, adjective, and others**. For token annotation, we use each model’s native tokenizer. During preprocessing, we remove special tokens (e.g., CLIP start/end tokens) and clean redundant symbols (e.g., underscores, extra spaces). Each token is aligned with the part-of-speech label of its corresponding word. Based on this pipeline, we construct global annotation files for training and test sets, shared across all layers to ensure fairness and reproducibility.
> > >
> > > **We adopt two probe architectures under a controlled-variable setup.** **For the MLP probe**, only the number of layers varies; the input dimension matches the text feature dimension, the hidden size is 128, each layer is Linear+ReLU+Dropout (0.2), and the output predicts five categories. **For the Transformer probe**, only the number of encoder layers varies; other settings are fixed (model dim 128, 8 heads, FFN dim 512), with input projection, Transformer encoder stack, and classification head, using ReLU and dropout 0.1. All probes are trained with Adam (lr $1\times10^{-4}$, weight decay $1\times10^{-5}$, batch size 64, 50 epochs), with a fixed random seed for reproducibility.
> > >
> > > In the main manuscript, we use a single-layer MLP probe and evaluate all models on randomly sampled timesteps with a fixed seed. The multi-timestep probing experiments reported in our previous response are conducted on SD3.5-Large under identical settings. These details will be added to the final manuscript.
> > >
> > > > **Response to W2&Q3**
> > >
> > > Our method's anchoring & restoration module has some limitations when handling cases like Qwen-Image, where the numerical scale of the target layer’s feature values is much larger than that of the origin layer. Future work can explore more robust alignment methods to address mismatches between features across layers. We provide the detailed mathematical derivation:
> > >
> > > In our **anchoring & restoration** method, we begin by normalizing both the origin and target features using LayerNorm:
> > >
> > > $$
> > > \hat{x}\_{ori} = LN(x\_{ori}), \hat{x}\_{tgt} = LN(x\_{tgt})
> > > $$
> > >
> > > Then, the origin features are injected into the target features with a scaling factor $\lambda$, and in the **restoration** step, we restore the features to the target's original scale using the target's mean $\mu_{tgt}$ and standard deviation $\sigma_{tgt}$:
> > >
> > > $$
> > > y = \big(\hat{x}\_{tgt} + \lambda \hat{x}\_{ori}\big) \odot \sigma\_{tgt} + \mu\_{tgt}
> > > $$
> > >
> > > Which can be split into:
> > >
> > > $$
> > > y = \hat{x}\_{tgt} \odot \sigma\_{tgt} + \mu\_{tgt} + \lambda \hat{x}\_{ori} \odot \sigma\_{tgt}
> > > $$
> > >
> > > **In the case where the target layer has a significantly larger mean**, i.e.,
> > >
> > > $$
> > > \|\mu_{tgt}\| \gg \|\lambda \sigma_{tgt} \hat{x}_{ori}\|
> > > $$
> > >
> > > the injected origin semantics only cause a **small perturbation** to the target layer's feature. This results in a minimal impact of the origin features on the final output.
> > >
> > > > **Response to Q4**
> > >
> > > These minor declines may be caused by metric mismatch: our method mainly improves fine-grained compositional correctness, while the four visual quality metrics capture partially different properties, such as coarse semantic alignment, aesthetics, or human preference. Notably, the observed decreases (i.e., FLUX PickScore and Qwen-Image CLIP) are very small and remain close to the boundary of the 95% confidence intervals, indicating that they do not constitute obvious performance degradation.
> > >
> > > **For CLIP**, the slight decrease may be due to metric mismatch. CLIP mainly captures global text–image semantic similarity and is less sensitive to fine-grained improvements such as attribute binding, counting, and spatial relations.
> > >
> > > **For PickScore**, the slight decrease may reflect its focus on pairwise human preference rather than strict prompt fidelity. In particular, for compositional prompts involving multiple objects, especially when the combinations are uncommon or less natural, base models may implicitly simplify the scene by omitting or deemphasizing certain elements to maintain visual coherence. With prompt reinjection, our method enforces stricter adherence to all prompt elements, making all objects explicitly present. While this improves compositional correctness, it can also lead to visually less natural images, which may be less preferred in pairwise comparisons, resulting in a small drop in PickScore. We provide additional qualitative results to further support this analysis: https://anonymous.4open.science/r/figs-3856/.
> > >
> > >
> > > We sincerely hope to address all of the reviewer’s concerns, which have helped us gain a deeper understanding of our own research.

---

### Official Review · Reviewer_fmRR · 2026-03-07

**Soundness:** 3
**Presentation:** 3
**Significance:** 3
**Originality:** 3
**Overall Recommendation:** 4
**Confidence:** 4

**Summary:**

This paper identifies a "prompt forgetting" phenomenon in Multimodal Diffusion Transformers (MMDiTs) where text representations lose semantic detail as they move through deeper layers. The authors show this through layer-wise analysis using CKNNA, PCA visualization, and linear probing for linguistic attributes across models like SD3.5 and FLUX. To fix this, they propose Prompt Reinjection, a training-free method that takes aligned text features from shallow layers and adds them back into deeper blocks during inference. Testing on benchmarks like GenEval and DPG-Bench shows that this approach improves instruction following, especially for spatial relations and counting, without hurting image quality.

**Compliance With Llm Reviewing Policy:**

Affirmed.

**Final Justification:**

I tend to keep my original rating as accepting this paper.

**Key Questions For Authors:**

Have you tested if reinjecting features from multiple shallow layers (instead of just one origin layer) works better?

**Limitations:**

The authors do not discuss the imitations and potential negative societal impact of their work.

**Strengths And Weaknesses:**

Strengths：
1. The paper clearly identifies a specific problem in MMDiTs where the text branch drifts because it lacks direct supervision.
2. The analysis uses several different methods—geometry, distribution, and functional probing—to prove that information is actually lost.
3. The proposed solution is training-free and easy to add to existing models during the inference stage.It includes a wide range of experiments on the latest popular models like SD3.5 and FLUX.1-Dev.
4. The method shows consistent gains across multiple benchmarks and human preference metrics.

Weaknesses:
1. The performance of the method depends on finding the right "origin layer" and "injection weight," which might require manual tuning for every new model.
2. The improvement on the Qwen-Image model is smaller compared to other models, suggesting it might not scale perfectly to much larger parameters.
3. The paper does not compare the method against more complex training-based solutions or different text-branch architectures.

---

> ### Author Rebuttal · Authors · 2026-03-31
>
> > **Response to Weakness 1**
>
> Thank you for the comment. In practice, the two key hyperparameters of our method are stable and do not require careful per-model tuning: **the best origin layer consistently lies in the shallow region (typically layer 1 or 2), and w = 0.025 is the most reliable choice**.
>
> As shown in our additional experiments in the response to **Reviewer 3t3u - Question 1**, this pattern remains stable across different benchmarks and CFG values. We observe the same trend on SD3, FLUX.1-dev, and SD3.5. In practice, using early layers (e.g., layer 1–2) with a small weight (w = 0.025) is sufficient to achieve robust gains, which we will clarify in the revision. We further validate this setting on a larger model (HunyuanImage-2.1), where it also yields clear improvements (see response to **Weakness 2**).
>
> ------
>
> > **Response to Weakness 2**
>
> Thank you for this thoughtful comment. To address scalability to much larger models, we provide additional results on **HunyuanImage-2.1 (17B)**. Using **origin = 1, w = 0.025** without exhaustive tuning, our method still yields clear gains on both **GenEval** and **DPG-Bench**:
>
> **GenEval**
>
> | Model            | Overall    | Single obj. | Two obj.   | Counting   | Colors     | Color attr | Position   |
> | ---------------- | ---------- | ----------- | ---------- | ---------- | ---------- | ---------- | ---------- |
> | HunyuanImage-2.1 | 0.7708     | 0.9906      | 0.9116     | 0.5906     | 0.8697     | 0.6475     | 0.6150     |
> | + Ours           | **0.8305** | **1.0000**  | **0.9394** | **0.7438** | **0.9149** | **0.6775** | **0.7075** |
>
> **DPG-Bench**
>
> | Model            | Overall     | Other       | Attribute   | Entity      | Global      | Relation    |
> | ---------------- | ----------- | ----------- | ----------- | ----------- | ----------- | ----------- |
> | HunyuanImage-2.1 | 85.4419     | 82.4000     | 89.2409     | 91.8901     | 81.7629     | **94.3520** |
> | + Ours           | **86.3294** | **84.0000** | **89.7390** | **92.4233** | **82.9787** | 94.1973     |
>
> **These results suggest that the method remains effective on much larger MMDiT backbones, and the smaller gain on Qwen-Image should not be taken as evidence of poor scalability.**
>
> For smaller improvement of Qwen-Image, we observed unusual early-layer feature dynamics. **Before layer 30, feature magnitudes grow rapidly from approximately 0–1 to the fp16 limit**, then remain near saturation due to explicit clipping (or around 6e5 in bf16/fp32). **We did not observe such severe numerical instability in the other models.**
>
> Accordingly, Qwen-Image appears to be a special case: **before layer 30, its features are unstable and do not show the same progressive clustering pattern as other backbones**, whereas from layer 30 to 60 they exhibit a more similar trend. We therefore apply our method in the 30–60 range for Qwen-Image. Although the gain is smaller, it remains positive, suggesting that prompt forgetting and prompt reinjection still hold in this setting.
>
> We will revise the paper to clarify both points.
>
> ------
>
> > **Response to Weakness 3**
>
> We thank the reviewer for this suggestion and added comparisons with two representative training-based methods. Compared with **TACA**, our method achieves stronger overall performance and more consistent gains across key instruction-following dimensions; detailed results are provided in **Appendix Table 12**. We also compared with **TAFS-GRPO**:
>
> | Model          | Overall |
> | -------------- | ------- |
> | FLUX           | 0.6613  |
> | FLUX+TAFS-GRPO | 0.8652  |
> | FLUX+Ours      | 0.6986  |
>
> While **TAFS-GRPO** achieves a much higher GenEval score, it is a heavy RL-based training method. In contrast, our method is **training-free and plug-and-play**, and its main value lies in directly revealing and mitigating the structural information-loss issue in MMDiT through a lightweight intervention. We will add these comparisons and clarify this distinction in the revised paper.
>
> ------
>
> > **Response to Question 1**
>
> Thank you for this insightful suggestion. We tested multi-layer reinjection by averaging features from multiple shallow layers as the source. The results show only a marginal gain over the best single-layer setting when averaging the first two layers, while including more layers degrades performance:
>
> | Source layers | GenEval Overall |
> | ------------- | --------------- |
> | origin-1      | 0.7059          |
> | origin-1–2    | **0.7069**      |
> | origin-1–3    | 0.6982          |
> | origin-1–4    | 0.6909          |
>
> This may be because using more shallow layers reduces the number of later blocks that receive reinjection.  We will include this comparison in the final version.
>
> > **Response to Limitation 1**
>
> We thank the reviewer for raising this important point. Our work is not intended to support imitation or other harmful uses, and we will add a brief discussion of these limitations and potential societal risks in the revised manuscript.
>
> ------

---

> > ### Author Rebuttal · Reviewer_fmRR · 2026-04-04
> >
> > N/A

---

> > > ### Author Response · Authors · 2026-04-04
> > >
> > > Thank you for your thoughtful feedback. We greatly appreciate your recognition of our efforts in addressing the major concerns raised. Your constructive comments have been very helpful in refining our work, and we are grateful for your time and consideration.

---

### Official Review · Reviewer_3t3u · 2026-03-10

**Soundness:** 3
**Presentation:** 3
**Significance:** 2
**Originality:** 2
**Overall Recommendation:** 4
**Confidence:** 2

**Summary:**

This paper studies instruction-following failures in multimodal diffusion transformers (MMDiTs), focusing on a phenomenon the authors call prompt forgetting. The authors argue that, because supervision is applied primarily on the visual branch during denoising, text representations in the text branch progressively lose fine-grained semantic information as depth increases. To support this claim, the paper analyzes intermediate text features in several representative MMDiT models, including SD3, SD3.5, FLUX, and Qwen-Image, using CKNNA, PCA-based visualization, and layer-wise probing of token-level semantic attributes. The authors appear to investigate a core question: whether text features in MMDiTs remain reliable conditioning signals throughout the denoising stack, or whether they degrade in a way that hurts instruction following.

Motivated by this analysis, the paper proposes Prompt Reinjection, a training-free inference-time intervention that reinjects shallow-layer text features into deeper transformer blocks. To reduce cross-layer mismatch, the method combines layer normalization-based distribution anchoring with orthogonal Procrustes alignment before reinjection. The method is evaluated on GenEval, DPG-Bench, and T2I-CompBench++, as well as several quality-oriented automatic metrics. The paper reports consistent improvements in instruction following across four MMDiT-based models, especially on spatial and compositional sub-tasks, while largely preserving image quality metrics. Overall, a significant problem considered by the manuscript is whether the internal text representations of modern MMDiT generators remain semantically faithful enough to support complex prompt adherence.

**Compliance With Llm Reviewing Policy:**

Affirmed.

**Final Justification:**

My concerns have been fully addressed. But Qwen-Image seems to be a problem that could not fit in current framework. And in the submission results, Qwen-Image results mainly improve a little, or even degrade. The author provides an explanation, but not satisfying. But I think the paper is still worth being accepted.

**Key Questions For Authors:**

1. How stable are the best origin layer and reinjection weight across prompt families, guidance scales, and model sizes?

2. Does prompt reinjection affect image diversity?

**Limitations:**

yes

**Strengths And Weaknesses:**

# Strength
- The paper provides a fairly coherent empirical argument for prompt forgetting. The combination of CKNNA, PCA-based drift analysis, and layer-wise probing is stronger than relying on a single diagnostic, and the monotonic decline in probing accuracy across depth is reasonably convincing as evidence that prompt-related information becomes less recoverable in deeper layers.
- The proposed method, Prompt Reinjection, is training-free and operates entirely at inference time by reinjecting shallow-layer text features into deeper transformer blocks. This design is appealing from an engineering standpoint because it avoids retraining large-scale diffusion models and can be applied to multiple existing MMDiT architectures.
- The paper is generally well organized. The motivation is clear, the method is easy to follow, and the figures are clear for illustration.

# Weakness
- The proposed method may be somewhat over-engineered relative to the final gain. The paper adds distribution anchoring, restoration, and orthogonal Procrustes alignment, yet it is not entirely clear whether these are fundamentally necessary or mostly helpful refinements.
- The benchmark improvements are consistent, but in many cases modest. For example, preference-style metrics and CLIP-based scores change only slightly.
- Limited novelty of the proposed intervention. While the analysis of prompt forgetting is interesting, the proposed solution is essentially a form of cross-layer feature reuse through residual injection. Similar ideas—such as feature reuse, shallow feature reinforcement, or cross-layer conditioning—are widely used in diffusion models.

---

> ### Author Rebuttal · Authors · 2026-03-31
>
> > **Response to Weakness1**
>
> Thank you for the insightful comment. We clarify that both modules are necessary and address distinct issues.
>
> Our core design—cross-layer prompt reinjection—already yields clear gains (Table 5), but introduces two problems: **distribution mismatch** (feature statistics drift) and **geometric mismatch** (feature space misalignment). We therefore use Distribution Anchoring & Restoration to stabilize statistics and Orthogonal Procrustes Alignment to correct misalignment.
>
> Updated ablations in Table 5 show that removing either component hurts performance, indicating complementary rather than redundant effects.
>
> | Anchor&Restore | Rotation | SD3-medium | FLUX.1-Dev |
> | :------------: | :------: | :--------: | :--------: |
> |       ×        |    ×     |   0.6849   |   0.6816   |
> |       ✓        |    ×     |   0.6897   |   0.6823   |
> |       ×        |    ✓     |   0.6910   |   0.6877   |
> |       ✓        |    ✓     |   0.7054   |   0.7002   |
>
>
>
> > **Response to Weakness2**
>
> These metrics reflect overall visual quality rather than fine-grained instruction following. We report them to show that our gains do not compromise fidelity. The results in Table 2 confirms no consistent negative trend, with most shifts being minor and positive. Thus, our method improves prompt adherence without sacrificing general image quality.
>
>
>
> > **Response to Weakness3**
>
> Our core contribution is uncovering and quantifying deep semantic decay in MMDiTs and we further providing a targeted, lightweight solution to alleviate it.
> We would appreciate specific pointers on "feature reuse" or "shallow feature reinforcement", "cross-layer conditioning." Despite an extensive search, we did not find specific prior work that directly corresponds to these concepts, and would appreciate concrete references to better contextualize our research.
>
>
>
> > **Response to Question1**
>
> We have conducted all the relevant experiments as suggested. Overall, shallow origin layers (specifically origin-1/2) and a moderate reinjection weight ($w=0.025$) consistently remain the most reliable choices across various settings.
> For prompt families, we repeated the hyperparameter search on DPG-Bench:
>
> | Origin Layer |  w=0.01 |     w=0.025 |  w=0.05 |   w=0.1 |
> | ------------ | ------: | ----------: | ------: | ------: |
> | origin-0     | 85.9436 |     86.9922 | 86.6919 | 72.6188 |
> | origin-1     | 85.9887 | **87.1396** | 86.9649 | 83.4499 |
> | origin-2     | 85.4213 |     85.9867 | 85.7642 | 80.1189 |
> | origin-4     | 85.8670 |     85.7130 | 85.7756 | 84.2060 |
> | origin-8     | 85.2935 |     86.2352 | 86.6493 | 85.4696 |
>
> For guidance scales, we searched under different CFG:
>
> | Origin Layer | w     |   CFG 4.0   |   CFG 6.0   |   CFG 7.0   |   CFG 8.0   |
> | :----------- | :---- | :---------: | :---------: | :---------: | :---------: |
> | origin-0     | 0.01  |   0.69020   |   0.68651   |   0.69010   |   0.68821   |
> |              | 0.025 |   0.69147   |   0.68991   |   0.68530   |   0.68762   |
> |              | 0.05  |   0.63613   |   0.65327   |   0.65930   |   0.64853   |
> |              | 0.1   |   0.23453   |   0.23872   |   0.24330   |   0.23214   |
> | origin-1     | 0.01  |   0.68481   |   0.68791   |   0.69460   |   0.68740   |
> |              | 0.025 | **0.69643** | **0.70201** | **0.70540** | **0.70108** |
> |              | 0.05  |   0.68447   |   0.69864   |   0.69670   |   0.69383   |
> |              | 0.1   |   0.52843   |   0.57417   |   0.58580   |   0.57243   |
> | origin-2     | 0.01  |   0.67471   |   0.69003   |   0.69190   |   0.68862   |
> |              | 0.025 |   0.68045   |   0.68489   |   0.70100   |   0.68499   |
> |              | 0.05  |   0.65993   |   0.67508   |   0.68140   |   0.66886   |
> |              | 0.1   |   0.50640   |   0.55210   |   0.55760   |   0.56157   |
> | origin-8     | 0.01  |   0.67128   |   0.68416   |   0.68730   |   0.68558   |
> |              | 0.025 |   0.66758   |   0.68569   |   0.68310   |   0.68266   |
> |              | 0.05  |   0.68169   |   0.68450   |   0.68170   |   0.67527   |
> |              | 0.1   |   0.62492   |   0.64224   |   0.64040   |   0.64311   |
>
> For model sizes, we searched on SD3-medium (2B) and FLUX.1-dev (12B) (Appendix Tables 10-11).
>
> ---
>
>
>
> > **Response to Question 2**
>
> We appreciate the insightful question. We evaluated image diversity using Vendi Score [1] on COCO5K (10 random seeds per prompt), calculated the mean Vendi Score with 95% confidence intervals.
> The results are shown below:
>
> | Method        | Mean Vendi Score | 95% CI   |
> | ------------- | ---------------- | -------- |
> | SD3-medium    | 2.0973           | ± 0.0338 |
> | + Reinjection | 2.0792           | ± 0.0322 |
>
> The slight decrease is statistically insignificant relative to the confidence intervals. This confirms that Reinjection does not materially compromise image diversity while significantly improving instruction following.
>
> [1] *The Vendi Score: A Diversity Evaluation Metric for Machine Learning*
>
> ------

---

> > ### Author Rebuttal · Reviewer_3t3u · 2026-04-01
> >
> > My concerns have been fully addressed. But Qwen-Image seems to be a problem that could not fit in current framework. And in the submission results, Qwen-Image results mainly improve a little, or even degrade. The author provides an explanation, but not satisfying. But I will raise the score, considering the supplement experiments.

---

> > > ### Author Response · Authors · 2026-04-04
> > >
> > > Thank you for your thoughtful comments. We are pleased to hear that our responses have addressed all your main concerns. Your feedback has played an important role in improving the quality of our work, and we sincerely appreciate your time and effort in reviewing it.
> > >
> > > > **For the concerns regarding Qwen-Image**
> > >
> > > Our method's anchoring & restoration module has some limitations when handling cases like Qwen-Image, where the numerical scale of the target layer’s feature values is much larger than that of the origin layer. Future work can explore more robust alignment methods to address mismatches between features across layers. We provide the detailed mathematical derivation:
> > >
> > > In our **anchoring & restoration** method, we begin by normalizing both the origin and target features using LayerNorm:
> > > $$
> > > \hat{x}\_{ori} = \mathrm{LN}(x\_{ori}), \qquad \hat{x}\_{tgt} = \mathrm{LN}(x\_{tgt})
> > > $$
> > > Then, the origin features are injected into the target features with a scaling factor $\lambda$, and in the **restoration** step, we restore the features to the target's original scale using the target's mean $\mu_{tgt}$ and standard deviation $\sigma_{tgt}$:
> > > $$
> > > y = \big(\hat{x}\_{tgt} + \lambda \hat{x}\_{ori}\big) \odot \sigma\_{tgt} + \mu\_{tgt}
> > > $$
> > > Which can be split into:
> > > $$
> > > y = \hat{x}\_{tgt} \odot \sigma\_{tgt} + \mu\_{tgt} + \lambda \hat{x}\_{ori} \odot \sigma\_{tgt}
> > > $$
> > > **In the case where the target layer has a significantly larger mean**, i.e.,
> > > $$
> > > \|\mu_{tgt}\| \gg \|\lambda \sigma_{tgt} \hat{x}_{ori}\|
> > > $$
> > > the injected origin semantics only cause a **small perturbation** to the target layer's feature. This results in a minimal impact of the origin features on the final output.

---

### Official Review · Reviewer_yHdP · 2026-03-12

**Soundness:** 3
**Presentation:** 3
**Significance:** 3
**Originality:** 3
**Overall Recommendation:** 4
**Confidence:** 4

**Summary:**

This paper studies prompt forgetting in MMDiT-based text-to-image models and proposes Prompt Reinjection, a training-free inference-time method that reinjects shallow text features into deeper layers. The paper argues that fine-grained prompt semantics become less recoverable in deeper layers, and shows that this simple intervention improves prompt adherence across several strong baselines.

**Compliance With Llm Reviewing Policy:**

Affirmed.

**Final Justification:**

I appreciate the authors’ effort in providing a detailed rebuttal. The responses have addressed my concerns, and I will raise my score to 4.

**Key Questions For Authors:**

Please see the weaknesses part.

**Limitations:**

yes

**Strengths And Weaknesses:**

Strengths
1. The paper is well motivated and addresses an important practical issue in modern text-to-image models.
2. The proposed method is simple. The empirical results are also promising, with consistent gains across multiple MMDiT backbones and benchmarks.

Weaknesses:
1. The prompts in COCO-5K are mostly short, everyday descriptions and do not reflect the kind of complex, long-form text prompts considered in this paper. I therefore question whether a rotation matrix calibrated on COCO-5K can reliably generalize to more complex and longer prompts.
2. The Procrustes rotation calibrated on COCO-5K appears largely heuristic. Since the cross-layer mapping is likely prompt-dependent and may vary across semantic categories and denoising stages, a single global orthogonal transform estimated from a generic caption dataset may be too coarse to accurately characterize the true geometry mismatch.
3. The method also appears to require careful model-specific tuning, including the selection of source/target layers and the recomputation of the alignment matrix for each backbone. This reduces the practical value of the claimed training-free setup, since the method is not truly plug-and-play across different models.

---

> ### Author Rebuttal · Authors · 2026-03-31
>
> > **Response to Weakness 1**
>
> We appreciate this comment and agree that a distribution gap exists. However, we clarify that our rotation matrix is computed via a global alignment on a large **token-level feature set aggregated from all calibration tokens**, **rather than aligning sentence-level embeddings**. This captures the intrinsic structure of the model’s text representation space, which is less tied to prompt length itself.
> To examine the reviewer's concern directly, we conducted an ablation study in **Table 9 (Appendix D)**. The results remain highly consistent across calibration sets with different scales and styles, suggesting that the learned mapping generalizes well as long as there is sufficiently diverse token coverage:
>
> | Calib. set     | C5K   | B1K   | B5K   | B10K  | E5K   |
> | -------------- | ----- | ----- | ----- | ----- | ----- |
> | GenEval (Ovr.) | 0.706 | 0.696 | 0.699 | 0.695 | 0.697 |
>
> Furthermore, **our superior performance on DPG-Bench (Table 3 & Appendix Fig. 7) directly validates our generalization to the long and challenging instruction-following prompts highlighted by the reviewer.**
>
>
>
> ---
>
>
>
> > **Response to Weakness 2**
>
> We agree that a single global rotation transform is an approximation. To address this, **we evaluated fine-grained, timestep-dependent alignment** on SD3-medium. We partition the denoising trajectory from 0 to 1000 into equal intervals of different granularity and compute a distinct rotation matrix for each, using features collected within that specific range. We then evaluated these variants on GenEval:
>
> | Setting           | GenEval Overall |
> | ----------------- | --------------- |
> | SD3-medium        | 0.6793          |
> | T1 (no partition) | 0.7054          |
> | T3 (3 intervals)  | 0.7086          |
> | T5 (5 intervals)  | 0.7116          |
>
> **Results confirm the reviewer’s intuition: performance scales with partition granularity (T5 > T3 > T1)**. This indicates that while our mixed-timestep strategy (T1) yields a robust “average” mapping by aggregating diverse denoising states, **a more dynamic treatment captures the evolving geometry mismatch more precisely**. We will incorporate this setup and analysis in the final version, and we thank the reviewer for this constructive suggestion, which provides an important direction for improving the method.
>
>
>
> ---
>
>
>
> > **Response to Weakness3**
>
> Thank you for this important comment.
>
> The key hyperparameters of our method are stable across backbones. As shown in **Tables 4, 10, and 11**, the best **origin** layer consistently lies in the shallow region (typically layer 1 or 2), and a moderate **reinjection weight** around 0.025–0.05 works reliably. In practice, these default choices already provide robust gains, while the small differences reported in the paper mainly reflect the best achievable setting for each backbone.
>
> The alignment matrix is also only a **one-time offline calibration** rather than training or adaptation. It can be computed within minutes, and we will release precomputed matrices for supported backbones. Together with the negligible inference overhead in **Table 7**, this keeps the method practical as a lightweight, training-free plug-in.
>
> To further verify the robustness of these hyperparameter choices, we additionally tested our method on the larger **HunyuanImage-2.1 (17B)** using a simple empirical configuration (**origin = 1, w = 0.025**) without tuning. The **GenEval overall score** improves from **0.7708** to **0.8305**, and the **DPG-Bench overall score** improves from **85.4419** to **86.3294**. Detailed subtask results are provided in our response to **Reviewer fmRR, Weakness 2**.
>
> We also clarify the special case in **Table 8**, where **Qwen-Image** shows a relatively deep best origin layer (origin = 30). For Qwen-Image, we observed unusual early-layer feature dynamics: before layer 30, feature magnitudes grow rapidly from approximately 0–1 to the fp16 limit, then remain near saturation due to explicit clipping (or around 6e5 in bf16/fp32). **We did not observe such severe numerical instability in the other models. **Accordingly, Qwen-Image appears to be a special case: **before layer 30, its features are unstable and do not show the same progressive clustering pattern as other backbones**, whereas from layer 30 to 60 they exhibit a more similar trend. We therefore apply our method in the 30–60 range for Qwen-Image. Although the gain is smaller, it remains positive, suggesting that prompt forgetting and prompt reinjection still hold in this setting.

---

> > ### Author Rebuttal · Reviewer_yHdP · 2026-04-03
> >
> > I appreciate the authors’ effort in providing a detailed rebuttal. The responses have addressed my concerns, and I will raise my score to 4.

---

> > > ### Author Response · Authors · 2026-04-04
> > >
> > > Thank you for your thoughtful review. We are glad to hear that the major concerns have been resolved to your satisfaction and greatly appreciate your recognition of our efforts. Your feedback has been crucial in enhancing the quality of our work, and we sincerely appreciate your time and effort in reviewing it.

---

### Decision · Program_Chairs · 2026-04-30

**Decision:**

Accept (regular)

**Comment:**

After the rebuttal, three reviewers vote for weak accept and one reviewer voted for weak rejection.

The decision is weighted in by checking the reviewer Za9G's comment and rebuttal.
For the question 1: Probing ≠ information loss, I partially agree with the reviewer that the declining in probing accuracy doesn't mean information loss. Even higher capability probe experiments don't reveal whether the information is truly loss or not. Actually I wonder whether there is any empirical way to examine this. I suggest the author to tune down this claim.
For the question regarding to Qwen-Image: I am leaning towards the author with the claim that Qwen-Image might be an outlier. The author should include those numerical analysis in the final version of this paper and the author should also include experiments of Hunyuan into the final version.

Given the above, I think the author might need to include those changes in the final version. I vote for weak accept.